# OmniBind: Large-scale Omni Multimodal Representation via Binding Spaces

**Zehan Wang**[*1,2]**, Ziang Zhang**[*1]**, Minjie Hong**[1]**, Hang Zhang**[1]**, Luping Liu**[3]**, Rongjie Huang**[1]**,
**Xize Cheng**[1]**, Shengpeng Ji**[1]**, Tao Jin**[1]**, Hengshuang Zhao**[2]**, Zhou Zhao**[†1,2]

[1]Zhejiang University; [3]Shanghai AI Lab; [3]The University of Hong Kong
`http://omnibind.github.io`

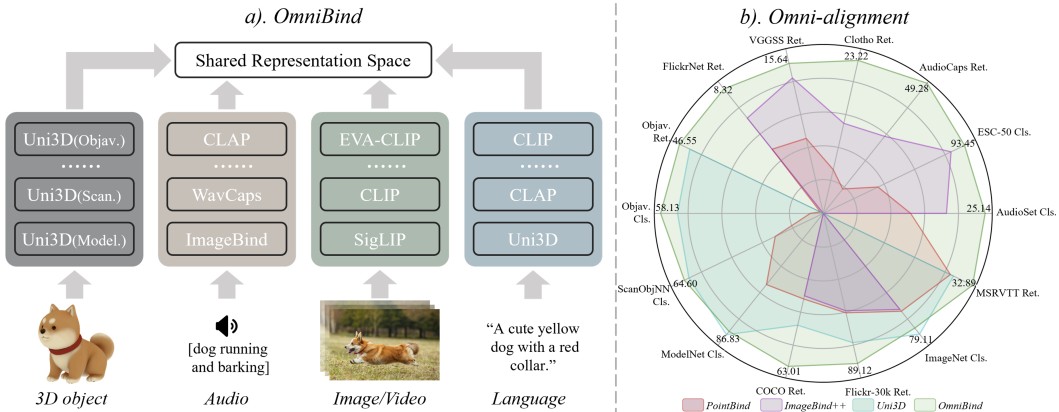

Figure 1: **Overview of OmniBind.** OmniBind integrates diverse knowledge of various existing multimodal models, leading to large-scale omni representations. OmniBind exhibits remarkable versatility and achieves state-of-the-art results on extensive downstream tasks over all modality pairs.

## Abstract

Recently, human-computer interaction with various modalities has shown promising applications, like GPT-4o and Gemini. Meanwhile, multimodal representation models have emerged as the foundation for these versatile multimodal understanding and generation pipeline. Models like CLIP, CLAP and ImageBind can map their specialized modalities into respective joint spaces. To construct a high-quality omni representation space that can be shared and expert in any modality, we propose to merge these advanced models into a unified space in scale. With this insight, we present **OmniBind**, advanced multimodal joint representation models via fusing knowledge of 14 pre-trained spaces, which support 3D, audio, image, video and language inputs. To alleviate the interference between different knowledge sources in integrated space, we dynamically assign weights to different spaces by learning routers with two objectives: cross-modal overall alignment and language representation decoupling. Notably, since binding and routing spaces only require lightweight networks, OmniBind is extremely training-efficient. Extensive experiments demonstrate the versatility of OmniBind as an omni representation model, highlighting its great potential for diverse applications, such as any-query and composable multimodal understanding.

## 1 Introduction

Multimodal joint representation, which aligns different modalities into a shared space, forms the foundation of current multimodal understanding (Liu et al., 2024a; 2023; Bai et al., 2023; Zhu et al., 2023b) and generation pipelines (Rombach et al., 2022; Podell et al., 2023; Singer et al., 2022; Huang et al., 2023b). Recently, co-understanding and generating various modalities with

---

[*]Equal Contribution.

[†]Corresponding author.

omni model has attracted increasing attention and demonstrates promising application prospects, like GPT-4o (OpenAI, 2024) and Gemini (Team et al., 2023).

Scaling up achieves incredibly success in language (Achiam et al., 2023; Touvron et al., 2023a; Jiang et al., 2024; Touvron et al., 2023b) and vision-language models (Alayrac et al., 2022; Liu et al., 2023; Sun et al., 2024). It consistently improves model performance and generalization by increasing the seen data and model parameters. In the field of multimodal representation models, some recent work scale models and data, but they are limited to certain combinations of modalities, such as image-text (Chen et al., 2023; Sun et al., 2024), video-text (Wang et al., 2022; 2024a), audio-text (Wu* et al., 2023; Mei et al., 2023), audio-image (Gong et al., 2022; Girdhar et al., 2023) and 3D-image-text (Zhou et al., 2023). Due to the lack of large-scale datasets covering various modalities, it is difficult to train omni multimodal representation models from scratch.

In this paper, we present **OmniBind**, a framework that binds advanced multimodal models, each excelling within their respective modalities (as depicted in Fig. 1-*a*). By integrating the alignment knowledge between different modalities, our approach achieves omni-alignment in an space shared by various modalities. Furthermore, by inheriting the pre-trained knowledge from these specialized models, our approach significantly lower the requirements of computational resources and training data. The entire training process can be completed with only several million unpaired data points using 4090 GPUs. The integration strategies and low resource demands makes our framework highly flexible and able to capitalize on improvements with any new pre-trained multimodal models.

However, it is non-trivial to effectively integrate numerous pre-trained spaces. Methods like remapping and weighted-averaging, as suggested in Wang et al. (2024b), fail to scale effectively when handling multiple source spaces. As more spaces are integrated, interference between the knowledge of different sources increases, leading to suboptimal performance. Manually adjusting the combining factors only results in trade-offs between different expertise rather than creating a truly versatile omni-model. We attribute this interference and the associated trade-offs to the rigidity of fixed weights. Since existing spaces are trained for different purposes using varied datasets, they encapsulate knowledge specific to certain aspects. Fixed-weight averaging tends to favor only particular aspects of knowledge, limiting the overall knowledge integration.

To unleash the potential of the integrated space that inherently contains all knowledge, we introduce a weight routing strategy designed to optimize the integration of different spaces. For each modality, we use a learnable router that dynamically determines the combining weights for different inputs. The training process of gating is guided by two primary goals: cross-modal overall alignment and language representation decoupling. The former motivates routers to predict optimal weights for all modality combinations, while the latter reduces conflicts among text embeddings aligned to different modalities, ensuring the clarity and distinction of language representations.

With the above techniques, we have successfully constructed three models that bind 5,13, and 14 spaces respectively. For 3D, audio, and image classification, our model exhibits advanced zero-shot generalization capabilities. Furthermore, it achieves significantly improved cross-modal alignment across all possible modality pairs compared to existing multimodal representations. This high-quality semantic omni-alignment enables advanced applications, including accurate 3D-audio retrieval, any-query object localization/audio separation, and complex compositional understanding.

Our contributions can be summarized as follows:

1) We propose **OmniBind**, a series of omni multimodal representation models that bind 5,13, and 14 spaces respectively and support five mainstream modalities: 3D point, audio, image, video, and language. It emphasizes the value of piecing various pre-trained specialist models together.

2) We introduce routers to ensemble spaces pre-trained on various modalities and datasets, thereby mitigating interference between knowledge from different sources and further enhancing versatility.

3) We design two learning objectives for learning routers: cross-modal overall alignment and language representation decoupling, which motivate routers to dynamically predict the optimal combining weights for all modality pairs while reserving the discrimination of representations.

4) OmniBind exhibits state-of-the-art performance on 14 benchmarks that cover all the modality pairs, and great potential for diverse applications, such as 3D-audio retrieval and any-query separation/localization, while requiring minimal training resources and data.

## 2 RELATED WORK

### 2.1 MULTIMODAL JOINT REPRESENTATION

Multimodal joint representation mainly aims to map inputs from different modalities into a shared space. Leveraging the semantic alignment property, pre-trained representation models are widely utilized in current multimodal large language models, such as LLaVA (Liu et al., 2024a; 2023), ImageBind-LLM (Han et al., 2023), and Chat-3D (Wang et al., 2023a; Huang et al., 2023a), as well as multimodal generation pipelines like Stable Diffusion (Rombach et al., 2022), SD XL (Podell et al., 2023), make-a-video (Singer et al., 2022) and make-an-audio (Huang et al., 2023b).

The initial multimodal representation is CLIP model (Radford et al., 2021), which demonstrates impressive generalization capabilities in various vision-language downstream tasks. Motivated by its success, successors propose stronger image-language representations by using higher-quality initialization (Fang et al., 2023c; Sun et al., 2023), larger-scale datasets (Schuhmann et al., 2022; Byeon et al., 2022), improved learning objectives (Zhai et al., 2023), or better model architecture (Fang et al., 2023b; Li et al., 2023). Besides, some researchers employ the multimodal contrastive learning paradigm in other modalities pair. LAION-CLAP (Wu* et al., 2023) and WavCaps (Mei et al., 2023) learn aligned audio-language space from audio-text pairs.

In addition to these studies improving the alignment of two modalities, another line of work aims to develop unified spaces capable of accommodating inputs from multiple modalities (more than two). For instance, AudioCLIP (Guzhov et al., 2022) and WAV2CLIP (Wu et al., 2022) introduce additional audio encoders for CLIP, leveraging audio-image-text and audio-image pairs. ULIP (Xue et al., 2023) and Uni3D (Zhou et al., 2023) collect massive amounts of 3D-image-text paired data, enabling them to learn 3D-image-text joint representations based on advanced image-text pre-trained models. More recently, ImageBind (Girdhar et al., 2023) and LanguageBind (Zhu et al., 2023a) propose to integrate multiple modalities using different data pairs that share the crucial image or language modality.

While current methods showcase a certain degree of robust cross-modal semantic alignment, they are primarily explored at relatively small scales and limited to specific modality pairs. On the other hand, our work focuses on exploring effective ways to bind a large number of omni representation spaces.

### 2.2 SCALING UP MODELS

Scaling up language or vision-language models has been incredibly successful in perception, reasoning, and generation tasks. GPT4 (Achiam et al., 2023) and LLaMA (Touvron et al., 2023a;b) achieve impressive conversation capability via training billion level language models on almost the massive internet language data. Recent InternVL (Chen et al., 2023) and EVA-CLIP-18B (Sun et al., 2024) try to scale the parameter of the CLIP model to the ten-billion level. These methods prove that with the growth of model parameters and seen data, models obtain consistent and non-saturating performance improvement.

However, most large-scale multimodal models are limited to aligning only two or three modalities. The scarcity of paired data across multiple modalities has hindered the development of large-scale omni multimodal representation models.

### 2.3 KNOWLEDGE FUSION

Fusing knowledge from different sources is a classical and wildly-used method to develop robust AI models. Traditional ensemble learning methods (Zhou & Zhou, 2021; Zounemat-Kermani et al., 2021) train models with different sub-datasets, and combine the output of different models as the final prediction. Similar ideas are also employed by large language model research, recent works (Bansal et al., 2024; Wan et al., 2024; Yu et al., 2023) propose to merge multiple language models tuned for different downstream tasks, and the resulting model excels at all aspects. Moreover, the Mixture-of-Experts (MoE) language model (Fedus et al., 2022; Jiang et al., 2024; Chowdhery et al., 2023) is also developing a hybrid model consisting of multiple sub-models and obtains better performance or efficiency by integrating them together.

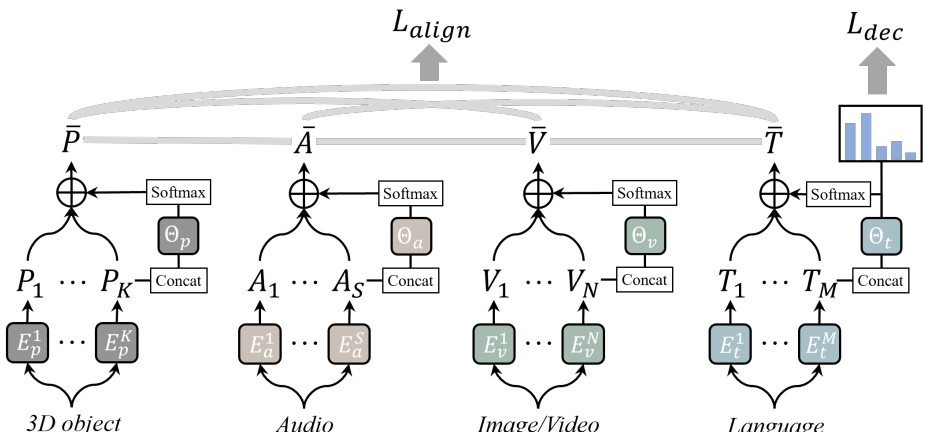

Figure 2: The pipeline of **OmniBind**. The $\Theta_X$ denotes the router of modality $X$, and $E_X^i$ is the $i$-th encoder of modality $X$. The losses $L_{align}$ and $L_{dec}$ are the objectives for training the routers.

For multimodal representation learning, C-MCR (Wang et al., 2023c) and Ex-MCR (Wang et al., 2023b) first propose to fuse two bi-modality representation space via the shared modality, thereby building unified space with low data and computing resource requirements. FreeBind (Wang et al., 2024b) is a high-level abstract of the above two methods, which employs bi-modality spaces to augment pre-trained unified space. Unlike these methods that only involve a few numbers of spaces, our goal is to obtain and keep improving omni representations via binding scale of spaces, which face more severe risks of knowledge interference.

## 3 METHOD

### 3.1 SPACES BINDING IN SCALE

Given that vision-language models are well-resourced and play a pivotal role in the multimodal field, we choose the advanced CLIP model EVA-CLIP-18B (Sun et al., 2024) as the foundation, and bind additional image-text, audio-text, audio-image-text and 3D-image-text spaces onto it.

For space binding, FreeBind (Wang et al., 2024b) firstly proposes to improve a representation space by integrating extra spaces. Its space binding pipeline can be summarized as two steps: 1) collecting pseudo embedding pairs across two spaces and 2) mapping one space to another. Our binding process is primarily derived from FreeBind, but we replace its pseudo embedding-pair aggregation with a more efficient and robust pseudo item-pair retrieval, and scale up integrated spaces.

Concretely, FreeBind first encodes massive unpaired unimodal data into embeddings of each space, and then uses the cross-modal similarity maps to aggregate pseudo embedding pairs across two spaces. The embedding pairs are unique for each pair of spaces. Therefore, when binding extensive spaces, repeatedly aggregating embeddings is needed, which is very resource-intensive. Besides, the non-shared pseudo pairs are also unstable due to the varying performance of existing spaces.

To bind spaces robustly and efficiently, we directly retrieve pseudo pairs across all modalities at the item level. Considering lots of unpaired 3D, audio, vision, and language data, and leveraging the most advanced 3D-image-text, audio-text, audio-image, and image-text retrieval model, we can take each modality as a starting point to retrieve the top-1 recall of data from other modalities. This approach constructs the pseudo item pairs $\{p, a, v, t\}$. For simplicity, letters $p$, $a$, $v$ and $t$ is correlated to point cloud, audio, image and text modality, respectively.

Using the pseudo data, we train simple projectors to individually bind each space to EVA-CLIP-18B. The training objective of the projector is multimodal contrastive loss between all pairs of involved modalities. For instance, when binding CLAP with EVA-CLIP-18B, the learning objective is:

$$L_{bind} = \text{Info}(\Psi(\mathbf{A}_{at}), \mathbf{T}_{vt}) + \text{Info}(\Psi(\mathbf{A}_{at}), \mathbf{V}_{vt}) + \text{Info}(\Psi(\mathbf{T}_{at}), \mathbf{T}_{vt}) + \text{Info}(\Psi(\mathbf{T}_{at}), \mathbf{V}_{vt}) \quad (1)$$

where $\mathbf{A}_{at}$, $\mathbf{T}_{at}$ are CLAP embeddings, $\mathbf{V}_{vt}$, $\mathbf{T}_{vt}$ are EVA-CLIP-18B embeddings and the $\Psi(\cdot)$ is an multi-layer perceptron projector. The $\text{Info}(\cdot, \cdot)$ is multimodal constrastive loss:

$$\text{Info}(\mathbf{X}, \mathbf{Y}) = -\frac{1}{2}\sum_{i=1}^{B}(\log \frac{e^{(\mathbf{x_i}\cdot\mathbf{y_i})/\tau}}{\sum_{j=1}^{N}e^{(\mathbf{x_i}\cdot\mathbf{y_j})/\tau}} + \log \frac{e^{(\mathbf{y_i}\cdot\mathbf{x_i})/\tau}}{\sum_{j=1}^{N}e^{(\mathbf{y_i}\cdot\mathbf{x_j})/\tau}}) \qquad (2)$$

Binding all different spaces together results in a hybrid model with $K$ 3D point encoders, $S$ audio encoders, $N$ image encoders, and $M$ text encoders. These encoders originate from different pre-trained models are sharing the same representation space after binding. The representations of the resulting ensemble model can be calculated as:

$$\bar{\mathbf{P}} = \sum_{i=1}^{K}\alpha_i\cdot\mathbf{P}_i; \quad \bar{\mathbf{A}} = \sum_{i=1}^{S}\beta_i\cdot\mathbf{A}_i; \quad \bar{\mathbf{V}} = \sum_{i=1}^{N}\gamma_i\cdot\mathbf{V}_i; \quad \bar{\mathbf{T}} = \sum_{i=1}^{M}\delta_i\cdot\mathbf{T}_i; \qquad (3)$$

where the $\bar{\mathbf{P}}$, $\bar{\mathbf{A}}$, $\bar{\mathbf{V}}$, $\bar{\mathbf{T}}$ are the embeddings of resulting model, and $\mathbf{P}_i$, $\mathbf{A}_i$, $\mathbf{V}_i$, $\mathbf{T}_i$ respectively denote the 3D point, audio, vision, and text representations from each corresponding $i$-th encoder. The $\alpha_i$, $\beta_i$, $\gamma_i$, $\delta_i$ are the combining factors for $i$-th encoder of each modality.

## 3.2 WEIGHTS ROUTING

Existing works (Wang et al., 2023c;b; 2024b) manually set the combining factors of encoders from different spaces. While manual settings offer flexibility to customize the resulting space when integrating a few spaces, it is increasingly complex and impractical as more spaces are added. Moreover, hand-designed combining weights also limit the deep integration across various knowledge sources, leading to simple trade-offs rather than comprehensive incorporation of different expertise.

To address this issue, drawing inspiration from the Mixture-of-Expert (MoE) technique in Large Language Models (LLMs), we propose dynamically assigning weights with learnable routers. As shown in Fig. 2, each modality contains one router to predict the corresponding combining factors for encoders of this modality, which can be formulated as:

$$\alpha_1,\ldots,\alpha_K = \text{softmax}(\Theta_p(\mathbf{P})); \; \beta_1,\ldots,\beta_S = \text{softmax}(\Theta_a(\mathbf{A}))$$
$$\gamma_1,\ldots,\gamma_N = \text{softmax}(\Theta_v(\mathbf{V})); \; \delta_1,\ldots,\delta_M = \text{softmax}(\Theta_t(\mathbf{T})) \qquad (4)$$

where $\mathbf{P}$, $\mathbf{A}$, $\mathbf{V}$, $\mathbf{T}$ are the concatenated outputs of each modality, and the $\Theta_p$, $\Theta_a$, $\Theta_v$, $\Theta_t$ denote the router for 3D point, audio, vision and language, respectively. To develop effective and robust routers, we utilize the entire retrieved pseudo dataset $\{p, a, v, t\}$ and design two learning objectives.

**Cross-modal Overall Alignment.** To motivate routers to predict the optimal weights for all modality combinations, we employ contrastive losses overall modality pairs as the first learning target:

$$L_{align} = \text{Info}(\bar{\mathbf{A}}, \bar{\mathbf{P}}) + \text{Info}(\bar{\mathbf{A}}, \bar{\mathbf{V}}) + \text{Info}(\bar{\mathbf{A}}, \bar{\mathbf{T}}) + \text{Info}(\bar{\mathbf{P}}, \bar{\mathbf{V}}) + \text{Info}(\bar{\mathbf{P}}, \bar{\mathbf{T}}) + \text{Info}(\bar{\mathbf{V}}, \bar{\mathbf{T}}) \quad (5)$$

where $\bar{\mathbf{P}}$, $\bar{\mathbf{A}}$, $\bar{\mathbf{V}}$, $\bar{\mathbf{T}}$ are defined in Eq. 3. By simply averaging the contrastive losses between all modality pairs, we cultivate balanced routers that achieve comprehensively high-quality cross-modal semantic alignment over all modalities.

**Language Representation Decoupling.** Compared to 3D points, audio, images, and videos, which are primarily sampled from the real world, language data is entirely artificial, exhibiting much higher information density and a stronger ideographic tendency. Therefore, textual descriptions of different modalities exhibit significant biases: image captions often describe appearances, audio captions focus on sounding actions, and 3D captions prioritize spatial structures. As a result, text encoders trained to align different modalities demonstrate more specialized expertise than encoders of other modalities.

Considering the significant distribution variance among the different text representations, we introduce an auxiliary learning objective for the language router to disentangle the language representation and improve its generalization. It preserves the discrimination of text embedding space and enhances the semantic alignments with various modalities. Specifically, we drive the language router to identify which modality the input texts are likely describing and to prioritize text encoders that are specialized in the corresponding modality. To this end, we define the loss function as follows:

$$L_{dec} = -\sum_{j=1}^{M}[y_j \log(\Theta_t(\mathbf{T})_j) + (1 - y_j)\log(1 - \Theta_t(\mathbf{T})_j)] \qquad (6)$$

where the $M$ is the number of text encoders. The texts in the retrieved pseudo-paired dataset are collected from audio-text, vision-text, and 3D-text pairs. Correspondingly, the text encoders are also derived from models pre-trained for audio-text, vision-text, and 3D-text alignment. In Eq. 6, $y_j = 1$ if the input text and the $j$-th text encoder are related to the same modality and $y_j = 0$ otherwise.

Finally, we linearly combine the above two objectives, and the final loss can be expressed as:

$$L = \lambda L_{dec} + L_{align} \qquad (7)$$

### 3.3 MODEL CONFIGURATIONS

The projectors $\Psi$ used for aligning spaces are simple two-layer MLPs. Additionally, we employ the mixture-of-projectors strategy following Wang et al. (2024b). The routers $\Theta$ are similarly designed as two-layer MLPs, with an extra sigmoid activation function at the end.

We select 14 pre-trained spaces for binding, which can be grouped into five audio-text (three Wav-Caps (Mei et al., 2023), two LAION-CLAPs (Wu* et al., 2023)), five image-text (EVA-CLIP-18B (Sun et al., 2024), EVA02-CLIP-E (Fang et al., 2023b) two SigLIPs (Zhai et al., 2023), DFN-ViT-H (Fang et al., 2023a)), three 3D-image-text (three Uni3Ds (Zhou et al., 2023)) and one audio-image-text (ImageBind (Girdhar et al., 2023)) spaces. After binding all spaces to EVA-CLIP-18B, we construct three configurations of OmniBind by combining different spaces: **OmniBind-Base**, **OmniBind-Large**, and **OmniBind-Full** with 5, 13, and 14 individual spaces. The encoder parameters for each modality and the specific spaces used in each variant are detailed in Appendix B.

## 4 EXPERIMENT

### 4.1 IMPLEMENTATION

**Datasets & Hyper-parameter.** To construct the pseudo-paired data, we collect unpaired 3D point, audio, vision, and text data from the training set of existing datasets. For 3D data, we use the 800k 3D point clouds from Objaverse (Deitke et al., 2023). The audio and image data come from AudioSet (Gemmeke et al., 2017) and ImageNet (Deng et al., 2009) respectively. The text data sources from three kinds of datasets: 3D-text (Liu et al., 2024b), visual-text (Lin et al., 2014; Sharma et al., 2018) and audio-text (Kim et al., 2019; Drossos et al., 2020) datasets. Based on these unpaired uni-modal data, we employ state-of-the-art audio-text (WavCaps (Mei et al., 2023)), image-text (EVA-CLIP-18B (Sun et al., 2024)), audio-image (ImageBind (Girdhar et al., 2023)) and 3D-image-text (Uni3D (Zhou et al., 2023)) models to retrieve the pseudo item pairs, as discussed in Sec. 3.1. The temperature factors in contrastive losses are 0.03, and the $\lambda$ in Eq. 7 is 3.

**Benchmarks & Baselines.** To comprehensively access the performance of our omni representations, we conduct quantitative experiments across 14 benchmarks covering 7 downstream tasks, as summarized in Tab. 1. In these benchmarks, we compare OmniBinds with three groups of previous multimodal representation models: 1) 3D-image-text models: Uni3D's pre-trained and three fine-tuned variants (Zhou et al., 2023). 2) audio-image-text models: C-MCR (Wang et al., 2023c), Language-Bind (Zhu et al., 2023a), ImageBind (Girdhar et al., 2023), ImageBind++ (Wang et al., 2024b) and InternVL$_{IB}$++ (Wang et al., 2024b). 3) 3D-audio-image-text models: Ex-MCR (Wang et al., 2023b) and Point-Bind (Guo et al., 2023). Moreover, we provide the best results on each benchmark achieved by the 14 source specialist spaces for reference, denoted as "Individual Best".

Table 1: Statistic for evaluation tasks and benchmarks.

| Task | Modality | Benchmarks | Items |
|---|---|---|---|
| Zero-shot Classification | Audio | AudioSet (Gemmeke et al., 2017) | 19,048 |
| | | ESC-50 (Piczak, 2015) | 400 |
| | Image | ImageNet-1K (Deng et al., 2009) | 50,000 |
| | 3D | Objaverse-LVIS (Deitke et al., 2023) | 46,832 |
| | | ScanObjNN (Uy et al., 2019) | 2,890 |
| | | ModelNet40 (Wu et al., 2015) | 2,468 |
| Cross-modal Retrieval | Audio-Text | AudioCaps (Kim et al., 2019) | 964 |
| | | Clotho (Drossos et al., 2020) | 1,045 |
| | Audio-Image | VGG-SS (Chen et al., 2021) | 5,158 |
| | | FlickrNet (Senocak et al., 2018) | 5,000 |
| | Image-Text | COCO (Lin et al., 2014) | 5,000 |
| | | Flickr-30K (Young et al., 2014) | 1,000 |
| | Text-Video | MSR-VTT (Xu et al., 2016) | 2,990 |
| | 3D-Image | Objaverse-LVIS (Deitke et al., 2023) | 46,205 |

Table 2: Cross-modal retrieval results. Best result is **bolded**, and second best result is underlined.

| Models | Audio-Text | | | | Audio-Image | | | | Image-Text | | | | Text-Video | | 3D-Image | |
| | AudioCaps | | Clotho | | VGG-SS | | FlickrNet | | COCO | | Flickr30K | | MSRVTT | | Objaverse | |
| | R@1 | R@5 | R@1 | R@5 | R@1 | R@5 | R@1 | R@5 | R@1 | R@5 | R@1 | R@5 | R@1 | R@5 | R@1 | R@5 |
|---|---|---|---|---|---|---|---|---|---|---|---|---|---|---|---|---|
| Uni3D | × | × | × | × | × | × | × | × | 59.51 | 81.45 | 87.85 | 97.55 | - | - | 43.57 | 67.61 |
| C-MCR | 15.76 | 41.37 | 8.37 | 24.86 | 1.94 | 7.69 | 1.39 | 5.97 | 16.67 | 37.04 | 34.16 | 63.64 | - | - | × | × |
| LanguageBind | 12.42 | 36.70 | 11.32 | 31.03 | 2.55 | 9.86 | 1.52 | 6.36 | 53.24 | 76.48 | 82.36 | 96.19 | 32.64 | 55.43 | × | × |
| ImageBind | 9.24 | 27.47 | 6.64 | 17.28 | 14.82 | 35.67 | 7.68 | 20.79 | 57.28 | 79.54 | 86.04 | 96.97 | 26.73 | 48.21 | × | × |
| ImageBind++ | 29.16 | 62.98 | 13.67 | 33.19 | 15.48 | **39.26** | 8.01 | 21.87 | 57.01 | 79.23 | 85.91 | 97.03 | - | - | × | × |
| InternVL$_{IB}$++ | 29.11 | 62.30 | 12.66 | 32.75 | 14.40 | 36.78 | 7.74 | 21.85 | 61.07 | 82.00 | 89.30 | 98.09 | - | - | × | × |
| Ex-MCR | 19.07 | 47.05 | 7.01 | 22.04 | 2.13 | 8.13 | 1.57 | 5.94 | 40.24 | 64.78 | 71.89 | 90.55 | - | - | 2.54 | 8.25 |
| PointBind | 9.24 | 27.47 | 6.64 | 17.28 | 14.82 | 35.67 | 7.68 | 20.79 | 57.28 | 79.54 | 86.04 | 96.97 | 26.73 | 48.21 | 5.86 | 14.59 |
| OmniBind-Base | 43.61 | 76.02 | 20.94 | 46.77 | 14.11 | 35.74 | 7.67 | 21.65 | 56.94 | 80.11 | 85.99 | 97.02 | 24.23 | 46.08 | 34.34 | 58.40 |
| OmniBind-Large | 47.89 | 79.75 | 23.07 | 49.67 | 14.14 | 36.07 | 7.86 | 21.72 | 60.08 | 82.35 | 87.20 | 97.40 | 30.05 | 52.91 | 46.09 | 69.11 |
| OmniBind-Full | **49.28** | **80.09** | **23.22** | 49.36 | **15.64** | 38.19 | **8.32** | **23.49** | **63.01** | **84.41** | 89.12 | **98.00** | **32.89** | 55.76 | **46.55** | **69.92** |
| Individual Best | 48.22 | 81.15 | 23.57 | 49.13 | 14.82 | 35.67 | 7.68 | 20.79 | 63.69 | 84.09 | 90.83 | 98.33 | 33.54 | 55.95 | 43.57 | 67.61 |

Table 3: Zero-shot classification results. Uni3D, Uni3D(Objav.), Uni3D(Scan.) and Uni3D(Model.) represent the pre-trained and three fine-tuned version of Uni3D-g, respectively.

| Model | Audio | | | Image | | 3D | | | | | |
| | AudioSet | ESC-50 | | ImageNet | | Objaverse | | ScanObjectNN | | ModelNet40 | |
| | mAP | Top1 | Top5 | Top1 | Top5 | Top1 | Top5 | Top1 | Top5 | Top1 | Top5 |
|---|---|---|---|---|---|---|---|---|---|---|---|
| Uni3D | × | × | × | 80.12 | 95.97 | 53.13 | 81.59 | 64.12 | 91.63 | 87.56 | **99.27** |
| Uni3D(Objav.) | × | × | × | 80.12 | 95.97 | 54.74 | 82.54 | 58.89 | 88.69 | 84.20 | 98.42 |
| Uni3D(Scan.) | × | × | × | 80.12 | 95.97 | 50.99 | 80.00 | **65.81** | 92.70 | 88.05 | 98.58 |
| Uni3D(Model.) | × | × | × | 80.12 | 95.97 | 51.21 | 80.14 | 64.43 | 90.66 | 88.05 | 99.11 |
| C-MCR | 11.15 | 70.35 | 96.70 | 24.50 | 52.44 | × | × | × | × | × | × |
| LanguageBind | 18.33 | **94.90** | 99.70 | 77.15 | 94.64 | × | × | × | × | × | × |
| ImageBind | 13.96 | 67.25 | 87.50 | 76.31 | 94.23 | × | × | × | × | × | × |
| ImageBind++ | 19.69 | 90.30 | 99.30 | 76.01 | 94.36 | × | × | × | × | × | × |
| InternVL$_{IB}$++ | 18.93 | 87.75 | 98.75 | **81.21** | **96.68** | × | × | × | × | × | × |
| Ex-MCR | 6.67 | 71.20 | 96.80 | 60.79 | 86.98 | 17.94 | 43.37 | 40.31 | 77.20 | 66.53 | 93.60 |
| PointBind | 13.96 | 67.25 | 87.50 | 76.13 | 94.22 | 13.83 | 30.34 | 55.05 | 86.89 | 76.18 | 97.04 |
| OmniBind-Base | 21.19 | 92.90 | 99.75 | 76.18 | 94.02 | 53.30 | 81.85 | 57.79 | 89.76 | 82.82 | 97.12 |
| OmniBind-Large | 25.57 | 93.25 | 99.80 | 78.87 | 95.32 | 53.97 | **82.90** | 64.67 | 94.15 | 86.55 | 99.03 |
| OmniBind-Full | **25.87** | 94.55 | **99.90** | 79.11 | 95.49 | **58.13** | 81.76 | 64.60 | **95.53** | 86.83 | 99.03 |
| Individual Best | 23.36 | 94.05 | 99.75 | 82.43 | 96.73 | 54.74 | 82.54 | 65.81 | 92.70 | 88.05 | 99.11 |

## 4.2 Performance Results

**Cross-modal Retrieval.**    As aforementioned, OmniBind aims to provide high-quality semantic alignment between all modality pairs. Therefore, we comprehensively assess the cross-modal retrieval performance across all possible modality pairs. The quantitative results for audio-text, audio-image, vision-text, and 3D-image retrieval are presented in Tab. 2.

Overall, OmniBind-Full and OmniBind-Large consistently outperform all previous methods across all benchmarks. Some prior approaches demonstrate competitive performance within their specific domains. For instance, ImageBind++ shows similarly strong audio-image alignment, and InternVL$_{IB}$++ displays comparable image-text capabilities, but they both fall short in other areas and lack support for 3D input. Compared to existing 3D-audio-image-text models like Ex-MCR and PointBind, all OmniBind variants exhibit substantial and comprehensive advantages across all combinations of modalities. These observations underscore the versatility and superiority of OmniBind.

Moreover, OmniBind-Full achieves similar performance to the "Individual Best", demonstrating that OmniBind successfully inherits and effectively integrates the expertise of various source spaces. Remarkably, OmniBind achieves even better audio-image and 3D-image alignment, showcasing the exciting cross-space knowledge transfer. By incorporating the high-quality image representations learned from image-text data, the audio-image and 3D-image alignment can also be improved.

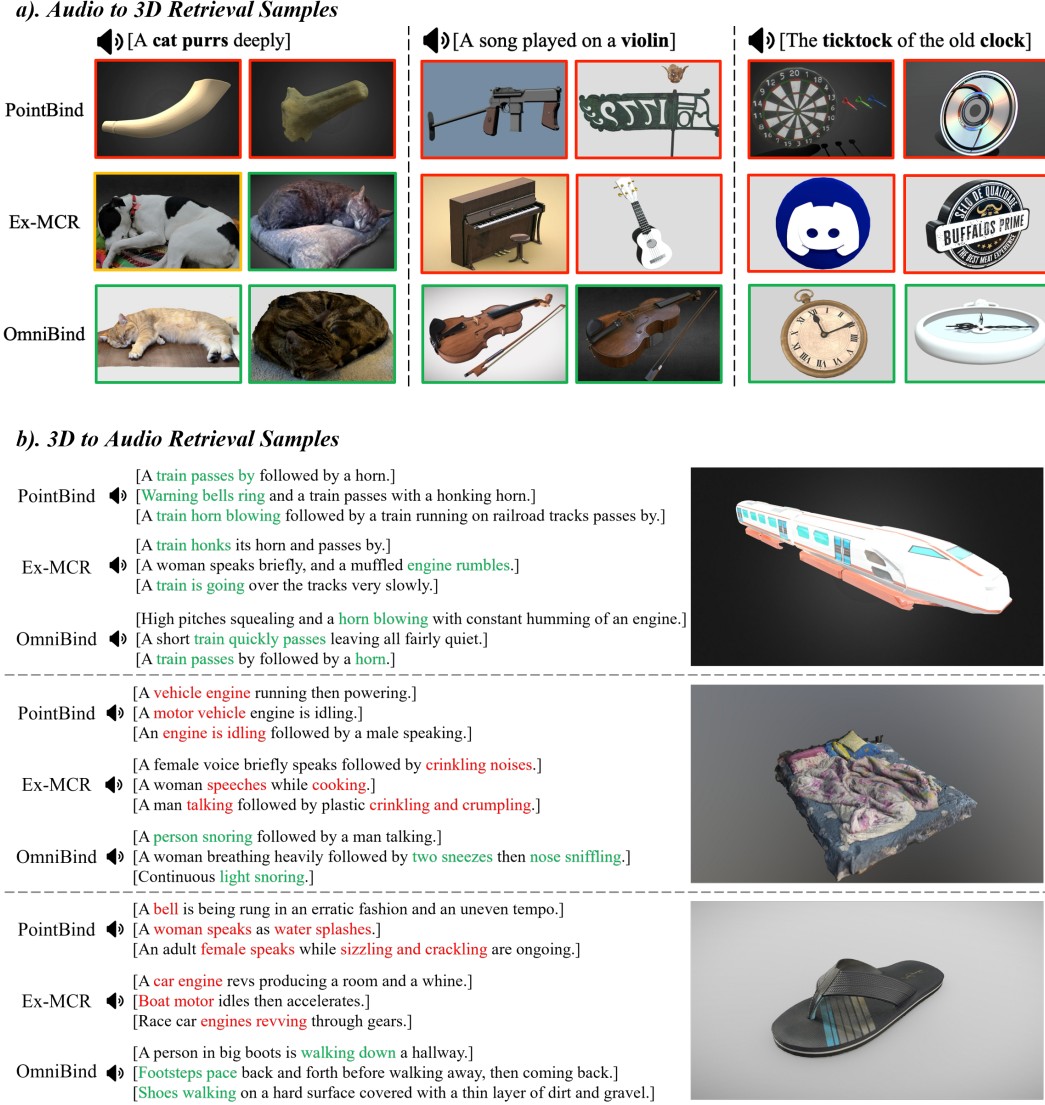

Figure 3: Qualitative comparison of audio-3D object retrieval. More visualizations are provided in the Appendix C.

**Zero-shot Classification.**  To further validate the generalization ability of OmniBind, we conduct zero-shot classification on each modality, and report the results in Tab. 3.

For audio and image classification, OmniBind demonstrates overall superiority, even when compared to models excelling in the audio-image-text domains. While LanguageBind performs slightly better than OmniBind on ESC-50, all variants of OmniBind significantly outperform previous methods on the more challenging AudioSet benchmark. This highlights the robustness and generalization of OmniBind. Furthermore, although InternVL$_{IB}$++ achieves the highest accuracy on ImageNet, it falls short in audio classification, which further showcases the versatility of OmniBind.

In 3D object classification, the three fine-tuned variants of Uni3D perform exceptionally well on their respective fine-tuned datasets. OmniBind-Base, which leverages only Uni3D(Objav.)  as the 3D-image-text source space, achieves performance comparable to Uni3D(Objav.), demonstrating that the binding space effectively inherits knowledge from the source space. Additionally, OmniBind-Large and OmniBind-Full integrate the knowledge from all three fine-tuned versions, performing well across various benchmarks and achieve significant improvements on the most challenging Objaverse-LVIS benchmark.

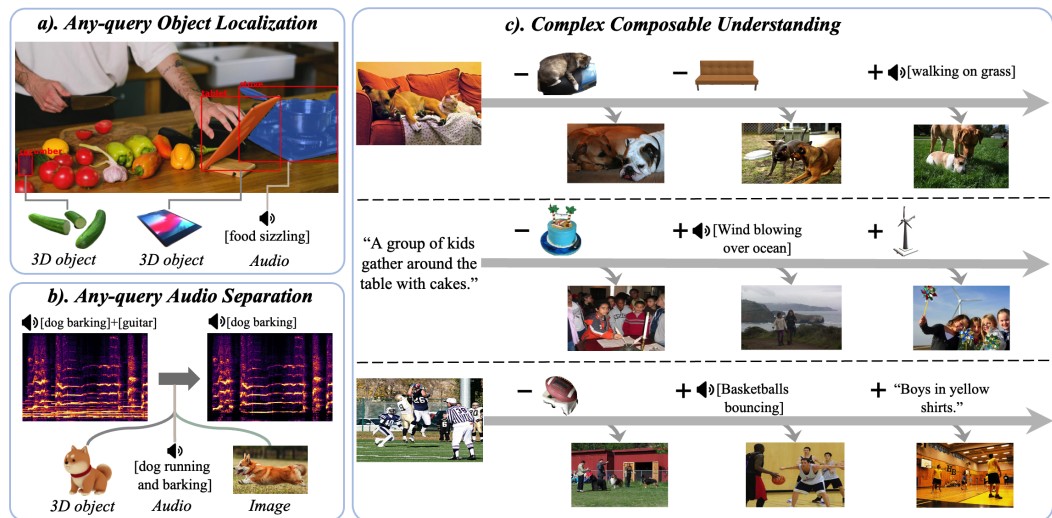

Figure 4: Diverse applications enabled by OmniBind. (a) Any-query Object Localization: accurately matches the object proposals with audio and 3D query. (b) Any-query Audio Separation: successfully separate audio using queries from images, audio, language, and 3D models. (c) Complex composable understanding. OmniBind space supports complex serial embedding arithmetic to freely add and remove semantic information.

## 4.3 APPLICATIONS

**Emergent cross-modal alignment.** OmniBind reveals a surprising emergent alignment between 3D and audio. In Fig. 3-a and 5, we randomly sample audios from AudioCaps and retrieve relevant 3D objects in Objaverse, and in Fig. 3-b, we randomly select 3D objects from Objaverse and retrieve relevant audio clips from AudioCaps. OmniBind shows a deep understanding of 3D and audio samples and high-quality semantic alignment. For example, it accurately recognizes the audio of "cat purring" and "violin" and finds the corresponding 3D models. In the "clock ticktock" case, while PointBind and Ex-MCR struggle to distinguish disk-shaped 3D models, OmniBind successfully identifies mechanical clocks.

**Any-query object localization.** OmniBind also demonstrates high-quality fine-grain semantic alignment, enabling any-query object localization. By extracting object proposals using pre-trained object segmentation (Kirillov et al., 2023) models, we can match these object proposals with queries from any modality within OmniBind's space. As shown in Fig. 4-a, we can localize objects in images using sounds or 3D models as queries.

**Any-query audio separation.** We replace the CLIP embeddings in CLIPSep (Dong et al., 2022) with OmniBind and fine-tune it on our pseudo dataset. In Fig. 4-b, we observe that we can successfully separate audios of a dog barking using queries from images, audio, language, and even 3D models of a dog.

**Complex composable understanding.** Impressively, OmniBind's embedding space not only enables addition and subtraction between arbitrary embeddings but also supports complex serial embedding arithmetic. Three representative cases are provided in Fig. 4-c. In the first case, starting with a photo of "one cat and one dog on sofa," we sequentially subtract the embeddings of a 3D cat and a 3D sofa to obtain "two dogs on sofa" and "two dogs not on sofa." Adding the sound of walking on grass then results in "two dogs on the grass." Similar exciting capabilities are observed in two other examples.

**More potentials.** Previous multimodel joint representations (Zhou et al., 2023; Girdhar et al., 2023; Zhu et al., 2023a) enable other applications, like point cloud painting (Zhou et al., 2023), any-to-any generation (Tang et al., 2024), and omnimulti modal LLMs (Han et al., 2023; Lin et al., 2023). Given that OmniBind is essentially homologous to the representation models used in previous applications and contains more diverse and generalizable knowledge, it holds great potential for these applications as well. We leave these explorations for future work.

Table 4: Ablation study of manual weights and the two weight routing objectives: $L_{align}$ and $L_{dec}$. *AT high*, *VT high*, and *PVT high* indicate manually assigning higher weights to audio-text, image-text, and 3D-image-text spaces, respectively. *Mean* means averaging all spaces. The Top1 and R@1 metrics of 3D classification and cross-modal retrieval are reported. The dataset names are abbreviated.

| Setting | $L_{align}$ | $L_{dec}$ | 3D classification | | | Audio-Text | | Audio-Image | | Image-Text | | 3D-Image |
|---|---|---|---|---|---|---|---|---|---|---|---|---|
| | | | Objav. | Scan. | Model. | ACaps | Clotho | VGGSS | FNet | COCO | F30K | Objav. |
| *AT high* | - | - | 49.47 | 59.90 | 85.17 | 44.47 | 21.76 | 14.33 | 7.69 | 55.97 | 84.81 | 46.23 |
| *VT high* | - | - | 53.44 | 64.19 | 86.99 | 33.24 | 17.49 | 13.82 | 7.45 | 61.38 | 88.57 | 46.24 |
| *PVT high* | - | - | 53.35 | 63.08 | 87.07 | 34.95 | 18.08 | 13.18 | 7.34 | 59.86 | 87.84 | **48.18** |
| *Mean* | - | - | 52.29 | 62.46 | 86.47 | 40.31 | 20.33 | 13.40 | 7.35 | 59.61 | 87.61 | 44.61 |
| Weight Routing | × | ✓ | 53.70 | **64.81** | **87.40** | 48.48 | **24.31** | 13.93 | 7.35 | 61.86 | 88.27 | 44.61 |
| | ✓ | × | 52.93 | 63.98 | 85.74 | 43.71 | 20.44 | 15.62 | **8.35** | 59.43 | 87.64 | 46.73 |
| | ✓ | ✓ | **58.13** | 64.60 | 86.83 | **49.28** | 23.22 | **15.64** | 8.32 | **63.01** | **89.12** | 46.55 |

## 4.4 ABLATION STUDY

**Performance improvement with more spaces.** The integrated spaces and parameters of our three variants: OmniBind-Base, OmniBind-Large, and OmniBind-Full, sequentially increase. From Tab. 2 and 3, we observe consistent overall performance improvements as binding more and more models. The scalability of the integrated spaces further highlight the potential of our method.

**Trade-offs in manually assigned weights.** Tab. 4 shows the performance of four types of manually assigned weights: *AT high*, *VT high*, *PVT high* and *Mean*. These four variants reveal trade-off phenomena among different areas of expertise. *AT high* performs well in the audio-text domain but is less effective in 3D classification and image-text tasks. Conversely, *VT high* (*PVT high*) obtains better image-text (3D-image) performance at the expense of audio-text alignment. *Mean* exhibits a more balanced performance but lacks specific expertise. Moreover, using weight routing showcases a comprehensive advantage over these manual weight variants. This observation further proves the routers can effectively alleviate interference between knowledge of different sources, leading to overall improvements in performance rather than simple trade-offs.

**Effectiveness of $L_{align}$ & $L_{dec}$ for learning routers.** We provide the ablation experiments for the two learning objectives in Tab. 4. By comparing the *Mean* and variants using each objective separately, we conclude that $L_{align}$ brings noticeable improvements across all possible modality combinations, while $L_{dec}$ specifically and significantly enhances language-related alignment. Combining the two objectives yields the best overall performance by complementing each other's strengths.

**Improved discrimination with $L_{dec}$.** To explore the effect of language representation decoupling on discrimination, we extract language embeddings from all AudioCaps and COCO captions. We find that employing $L_{dec}$ reduces the cosine similarity between all the possible language embedding pairs from 0.0828 to 0.0517. As discussed in (Liang et al., 2022), lower cosine similarity between irrelevant pairs indicates the embeddings "occupy" more space in the hypersphere. Therefore, employing $L_{dec}$ indeed enhances the discrimination of language representations, leading to better overall performance.

## 5 CONCLUSION

In this work, we present OmniBind, a series of omni multimodal representation models that bind 5,13, and 14 spaces. Our core contributions lie in designing the weight routing strategy to mitigate interference between knowledge of different sources, thereby successfully binding pre-trained spaces in scale to obtain a keep improving multimodal representation. By efficiently integrating multiple pre-training spaces, OmniBind demonstrates impressive multimodal performance across extensive benchmarks and shows vast potential for further growth and diverse applications.

## REPRODUCIBILITY STATEMENT

All checkpoints for the different versions of OmniBind will be open-sourced. The source spaces used to build OmniBind, as detailed in Table 8, along with the evaluation benchmarks in Table 1, are publicly accessible.

## ACKNOWLEDGEMENTS

This work was supported in part by the National Key R&D Program of China under Grant No.2022ZD0162000.

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

## A    LIMITATIONS AND FUTURE WORKS

Although OmniBind is already the largest multimodal representation model, demonstrating remarkable versatility and generalization, it currently utilizes only 14 existing spaces and 5 modalities: 3D, audio, images, video and text. It remains to be explored whether or when this binding space-based "scaling up" will reach saturation. Additionally, this work only presents three basic applications as inspiration. Investigating whether using higher-quality and larger-scale omni multimodal representations can enable new capabilities in downstream applications would be a promising direction.

## B    STATISTICS OF OMNIBIND VARIANTS

The detailed source spaces used to build OmniBind-Base, OmniBind-Large, and OmniBind-Full are presented in Tables 5, 6, and 7, respectively. Generally speaking, OmniBind-Base comprises three components: three WavCaps, EVA02-CLIP-E, Uni3D (Objaverse), and ImageBind. The primary difference between OmniBind-Large and OmniBind-Full is the inclusion of EVA-CLIP-18B in OmniBind-Full. Additionally, we provide statistics on the number of parameters for OmniBind-Base, OmniBind-Large, and OmniBind-Full in Table 8.

Table 5: Source spaces for building OmniBind-Base.

| OmniBind-Base | |
|---|---|
| Audio-Text | WavCaps (Mei et al., 2023), WavCaps (Clotho) (Mei et al., 2023), WavCaps (AudioCaps) (Mei et al., 2023) |
| Image-Text | EVA02-CLIP-E (Fang et al., 2023b) |
| 3D-Image-Text | Uni3D (Objav.) (Zhou et al., 2023) |
| Audio-Image-Text | ImageBind (Girdhar et al., 2023) |

Table 6: Source spaces for building OmniBind-Large.

| OmniBind-Large | |
|---|---|
| Audio-Text | WavCaps (Mei et al., 2023), WavCaps (Clotho) (Mei et al., 2023), WavCaps (AudioCaps) (Mei et al., 2023), LAION-CLAP (general) (Wu* et al., 2023), LAION-CLAP (music) (Wu* et al., 2023) |
| Image-Text | EVA02-CLIP-E (Fang et al., 2023b), DFN-ViT-H (Fang et al., 2023a), SigLIP-Large (Zhai et al., 2023), SigLIP-so400M (Zhai et al., 2023) |
| 3D-Image-Text | Uni3D (Objav.) (Zhou et al., 2023), Uni3D (Scan.) (Zhou et al., 2023), Uni3D (Model.) (Zhou et al., 2023) |
| Audio-Image-Text | ImageBind (Girdhar et al., 2023) |

Table 7: Source spaces for building OmniBind-Full.

| OmniBind-Full | |
|---|---|
| Audio-Text | WavCaps (Mei et al., 2023), WavCaps (Clotho) (Mei et al., 2023), WavCaps (AudioCaps) (Mei et al., 2023), LAION-CLAP (general) (Wu* et al., 2023), LAION-CLAP (music) (Wu* et al., 2023) |
| Image-Text | EVA-CLIP-18B (Sun et al., 2024), EVA02-CLIP-E (Fang et al., 2023b) DFN-ViT-H (Fang et al., 2023a), SigLIP-Large (Zhai et al., 2023), SigLIP-so400M (Zhai et al., 2023) |
| 3D-Image-Text | Uni3D (Objav.) (Zhou et al., 2023), Uni3D (Scan.) (Zhou et al., 2023), Uni3D (Model.) (Zhou et al., 2023) |
| Audio-Image-Text | ImageBind (Girdhar et al., 2023) |

## C    MORE VISUALIZATIONS

To further qualitatively assess the capabilities of OmniBind, we provide additional visualizations. Each sample is manually evaluated and categorized: correct samples are marked in green, incorrect ones in red, and partially correct ones in orange.

Furthermore, we present additional cases of audio to 3D object retrieval in Figs. 5. In each instance, the 3D objects retrieved by OmniBind exhibit significantly better semantic alignment with the audio query than those retrieved by Ex-MCR and PointBind.

## D    SOCIAL IMPACT

OmniBind is a method for learning large-scale multimodal representation models by aligning and reorganizing knowledge from existing pre-trained models. Therefore, the capability of OmniBind is

Table 8: Statistics about parameter number of three OmniBind variants.

| Variants | Parameter Number | | | | | |
|---|---|---|---|---|---|---|
| | 3D Encoder | Audio Encoder | Image Encoder | Text Encoder | Projector | Total |
| OmniBind-Base | 1.0B | 99M | 4.6B | 1.3B | 224M | 7.2B |
| OmniBind-Large | 3.0B | 329M | 6.0B | 2.5B | 431M | 12.3B |
| OmniBind-Full | 3.0B | 329M | 23.6B | 3.2B | 431M | 30.6B |

mainly inherited from the source pre-trained models. Implementing additional safety detection and filtering processes for these pre-trained models can effectively reduce the potential for misuse and mitigate negative social impacts of OmniBind.

## E   EXAMPLES OF PSEUDO PAIRS

In Fig. 6, 7, 8 and 9, we present the examples of pseudo pairs that retrieved from 3D objects, images, audio and language, respectively. There are two main findings: 1) The pseudo-pairs collected by state-of-the-art cross-modal retrieval models exhibit strong semantic consistency. 2) Treating different modalities as the strating point improves diversity and covers different semantic biases. The typical example is that there are obvious differences between the text descriptions retrieved from image (Figure.7) and audio (Figure.8).

## F   VISULIZATION OF LANGUAGE REPRESENTATIONS

We present a UMAP visualization of the language representations of audio captions (from Audio-Caps) and image captions (from COCO) in Fig. 10. Within the high-quality self-supervised language model, T5 Ni et al. (2021), the two kinds of captions reveal a clear domain gap, supporting the claim that textual descriptions for different modalities exhibit significant biases. Similarly, in OmniBind's representation space, a similar gap is observed, underscoring the discriminative capability of OmniBind's representations. Notably, while removing the language decoupling objective does not significantly alter the qualitative distribution map, quantitative results indicate that language decoupling reduces the representation similarity between audio and visual captions from 8.28 to 5.17, demonstrating improved discernibility and a more dispersed distribution.

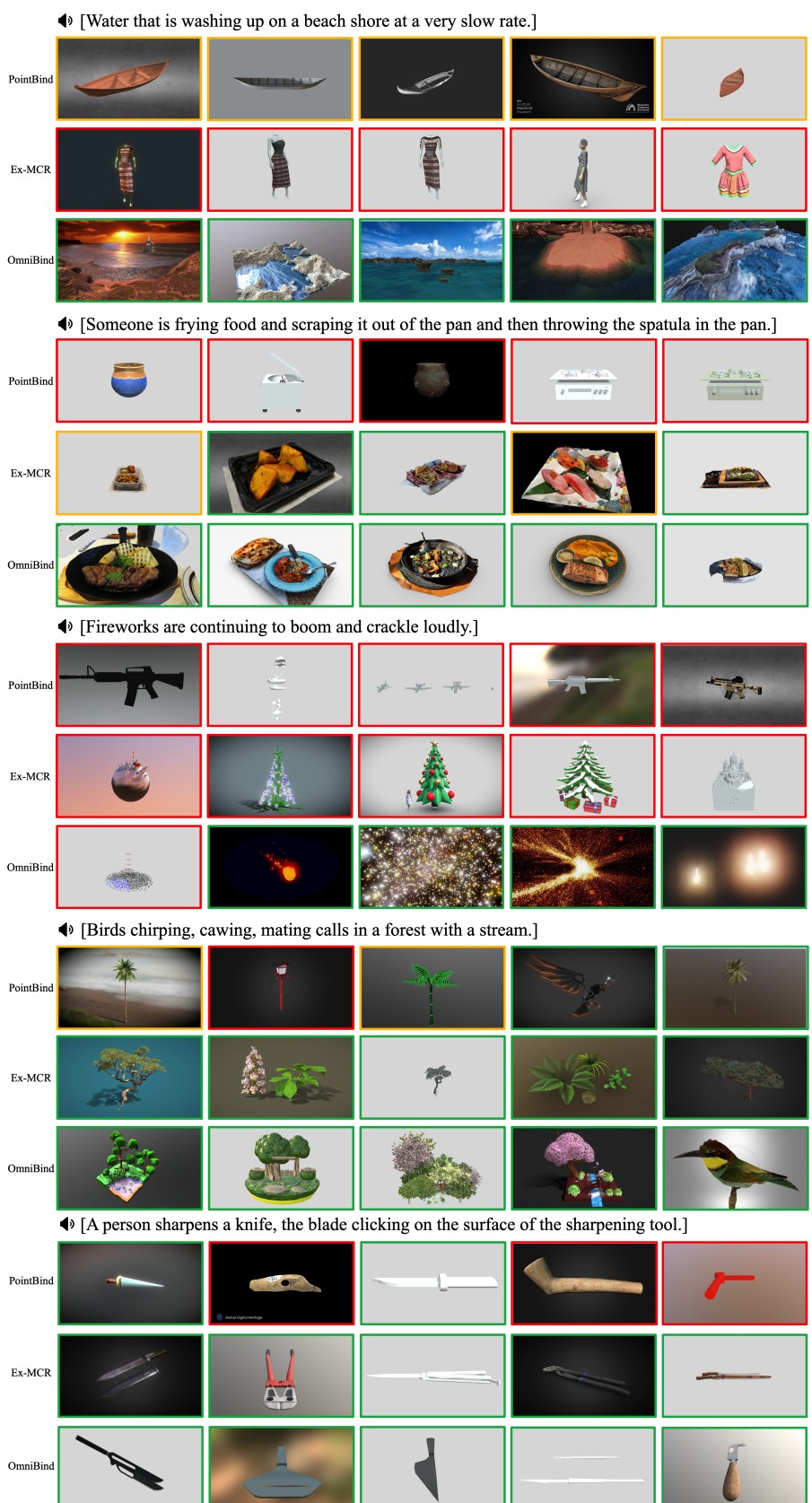

Figure 5: More visualization of Audio-to-3D retrieval.

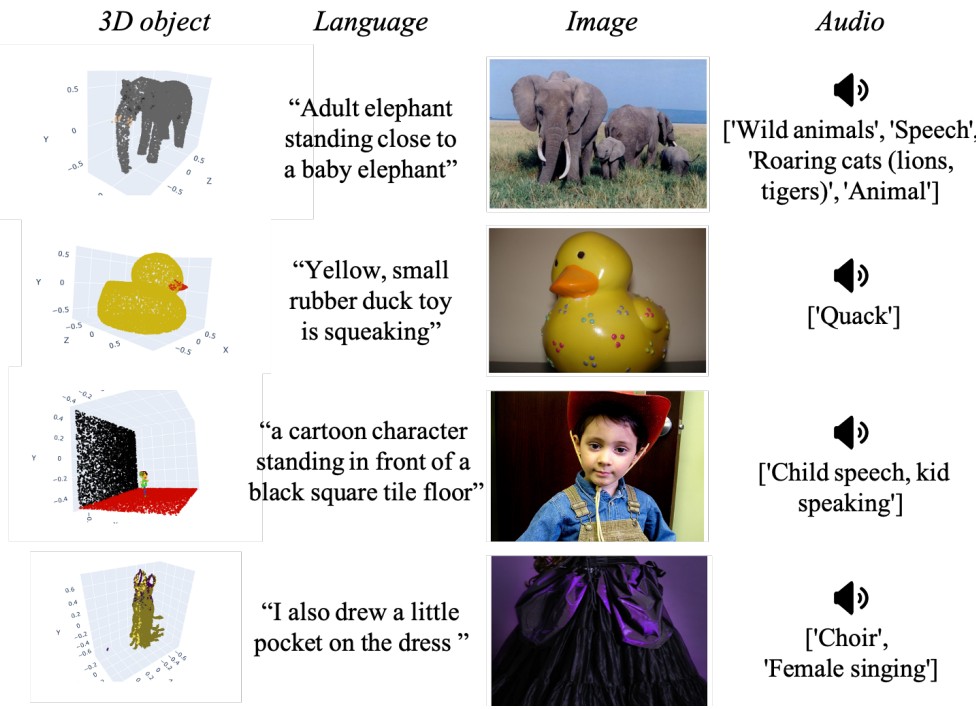

Figure 6: Examples of pseudo pairs retrieved from 3D objects.

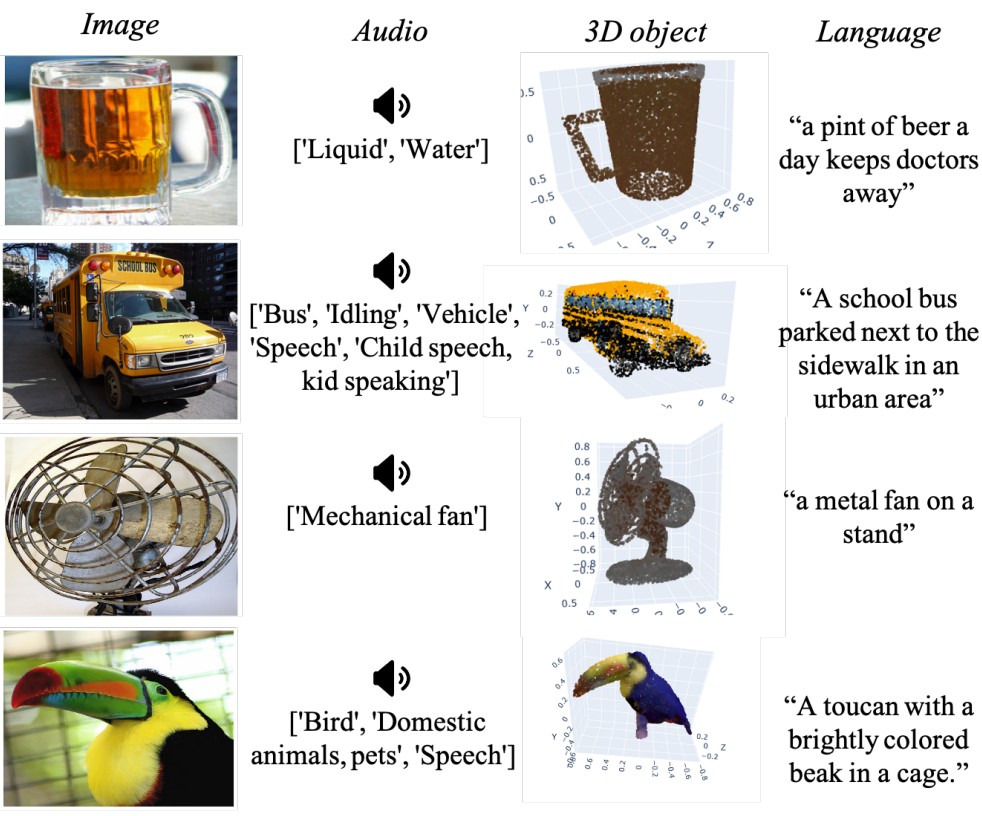

Figure 7: Examples of pseudo pairs retrieved from images.

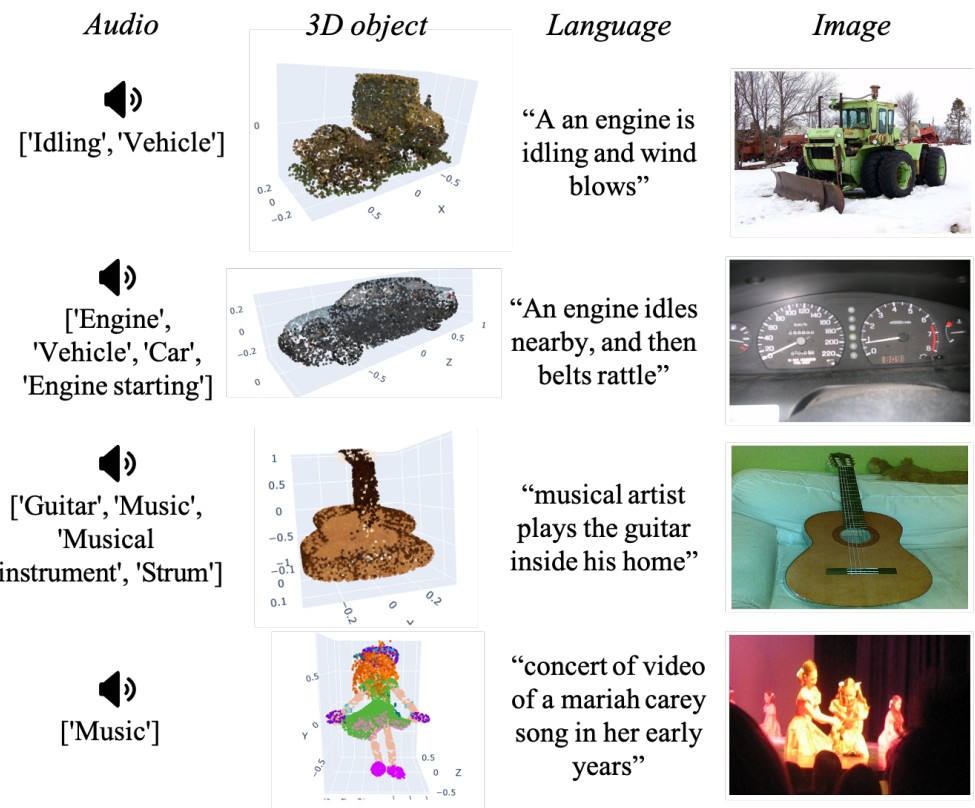

Figure 8: Examples of pseudo pairs retrieved from audios.

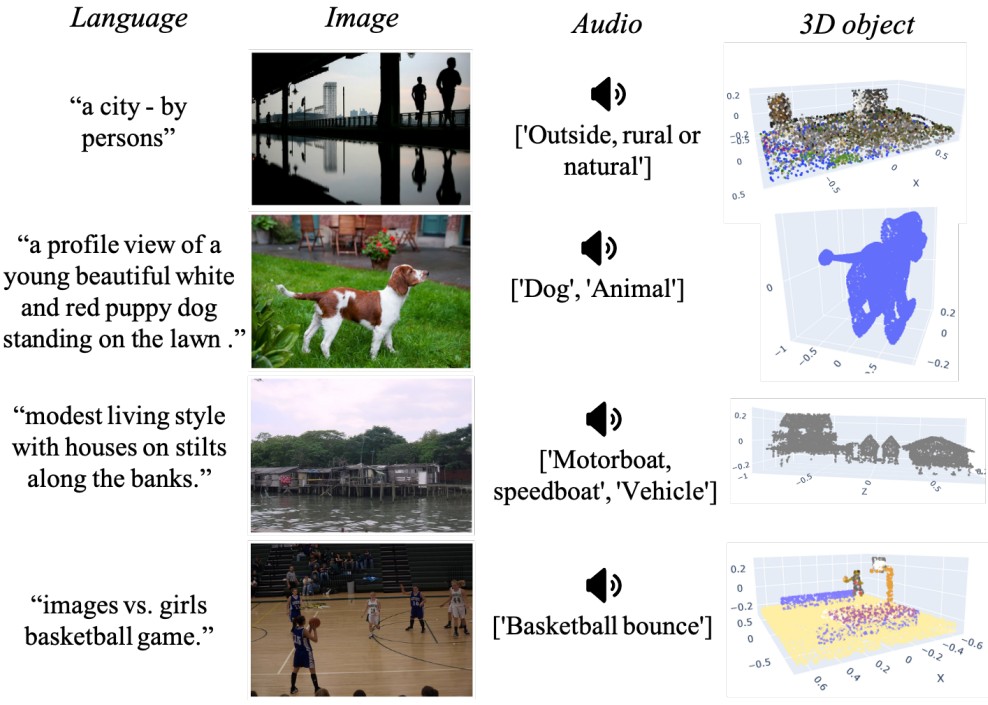

Figure 9: Examples of pseudo pairs retrieved from texts.

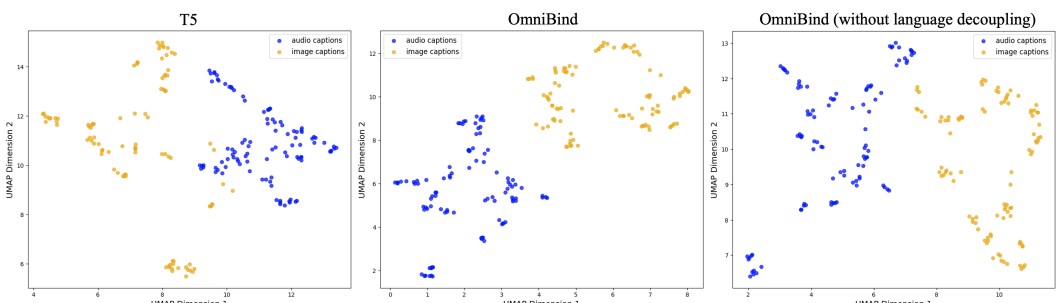

Figure 10: UMAP visualization of language representations from different domains in different pre-trained models.

