# OpenReview forum: "OmniBind: Large-scale Omni Multimodal Representation via Binding Spaces"
_ICLR.cc/2025/Conference — ICLR 2025 Poster_

### Official Review · Reviewer_enUq · 2024-11-03

**Soundness:** 2
**Presentation:** 2
**Contribution:** 2
**Rating:** 6
**Confidence:** 4

**Summary:**

This paper presents OmniBind, a framework designed to integrate diverse multimodal models into a unified representation space. This paper introduces a learnable router that dynamically assigns weights to different input combinations, mitigating interference between knowledge sources and enhancing versatility. By binding various pre-trained models, OmniBind achieves notable performance across multiple benchmarks and demonstrates versatility in handling different modalities.

**Strengths:**

1. This paper presents an approach OmniBind leverages the strengths of specialized models.
2. This paper addresses the limitation of fixed-weight averaging and offers a flexible way to integrate multiple knowledge sources.
3. This paper shows the potential of OmniBind for practical applications such as 3D-audio retrieval, any-query object localization, and complex multimodal compositional understanding.

**Weaknesses:**

1. This paper is somehow hard to follow and not well-written. The motivation and method sections may not be sufficiently clear. This paper contains some inconsistencies in notation and symbols, which is confusing for readers. For instance, the term "item pair {p, a, v, t}" on line 211 is not clearly defined or explained when it first appears.
2. This paper does not address how the introduction of OmniBind affects training and inference speeds.
3. This paper lacks detailed discussions on certain aspects of the method. For example, expert load balancing in the Mixture of Experts (MoE) has not been explored.
4. While the paper mentions that textual descriptions of different modalities exhibit significant biases (on line 258), it does not provide supporting evidence or references to illustrate these biases.
5. The generalizability to new modalities is not presented in this paper.

**Questions:**

1. This paper mentions in the abstract that OmniBind is extremely training-efficient (line 041). How is this reflected?
2. Did the paper consider the issue of expert load, and if so, how was it addressed?
3. On line 199 of the paper, the authors mention the advantages compared to FreeBind. How is item-pair retrieval done here?
4. If one wants to support a new modality under this method, how should it be carried out?

---

> ### Author Response · Authors · 2024-11-25
> **Responses to Reviewer enUq (1/n)**
>
> ### W1：Writing.
>
> We have added the explanation for the "item pair {p, a, v, t}" and will continue to refine the writing of the current manuscript.
>
>
>
> ### W2&Q1：Training and inference speeds.
>
> In terms of training speed, **our method only requires training the connector that binds spaces and the weight routers, both of which are lightweight MLP projectors. As a result, the training process can be efficiently completed using a single 4090 GPU and converges easily.** More specifically, on a single 4090 GPU, binding one space takes 5 hours, and learning the router for 14 models takes an additional 6 hours. Only a few million unpaired data are sufficient to achieve the current advanced performance. In contrast, training a multimodal representation model from scratch typically requires hundreds of times more computational resources and training data, as well as significantly longer training times.
>
>
>
> For inference, indeed, integrating multiple pre-trained models increases the inference latency of our models. According to the number of integrated encoders, OmniBind requires 2-5 times longer latency than previous methods in each modality. We propose OmniBind to explore an effective "scaling up" approach for multimodal representations, increasing model size for improved performance. **Scaling up models and boosting the current performance upper bound of multimodal representation models is a worthwhile pursuit. For example, large models can act as teacher models in distillation learning, where their knowledge is transferred to smaller models, leading to better models across all sizes, as seen in DINO v2 [1].** Proposing larger and more powerful multimodal representation models, therefore, holds substantial practical significance.
>
>
>
> [1] Oquab, Maxime, et al. "Dinov2: Learning robust visual features without supervision." arXiv preprint arXiv:2304.07193 (2023).
>
>
>
> ### W3& Q2：Expert load balancing.
>
> We intentionally avoid incorporating the expert load balancing mechanism used in the MoE structure of LLM, as the experts in MoE of LLM are generally homogeneous, while our encoders differ with prior biases (such as the text encoders from the audio-to-text model and the image-to-text model are experts in different fields). These inherent differences reduce the likelihood of relying excessively on a small subset of experts. Our experiments also empirically find that using the expert load balancing loss in MoE will lead to a decrease in the overall performance.
>
>
>
> |                          | Audio-text |           | Audio-Image |          | Image-Text |           | 3D-text   |           |           | 3D-Image  |
> | ------------------------ | ---------- | --------- | ----------- | -------- | ---------- | --------- | --------- | --------- | --------- | --------- |
> |                          | ACaps      | Clotho    | VGGSS       | FNet     | COCO       | F30K      | Objav.    | Scan.     | Model.    | Objav.    |
> | OmniBind-Full            | **49.28**  | **23.22** | **15.64**   | **8.32** | **63.01**  | **89.12** | **58.13** | **64.60** | 86.83     | **46.55** |
> | +  expert balancing loss | 47.84      | 23.07     | 14.16       | 7.42     | 57.86      | 86.33     | 52.63     | 63.56     | **86.75** | 46.20     |
>
>
>
> Additionally, we observe that our current routers can achieve a reasonable and dynamic balance between inputs from different domains, without using the expert load balancing loss, as shown in the router assign weight to Reviewer Z2z7’s Q3.
>
>
>
> ### W4：Biases between textual descriptions of different modalities.
>
> In the Figure.10 of the revised version of our paper, we visualize the representation distribution of texts from different domains in different spaces. In the well-studied self-supervised text representation space, sentence-T5, textual descriptions of audio and visual content show inherent domain gaps and occupy different subspaces.
>
>
>
> More analysis of the effectiveness of language decoupling can be found in the response to Reviewer ViaF’s W1.
>
> ### Q3：the advantages of item-pair retrieval.
> Please refer to the response to Reviewer qoeh’s W1.
>
> ### W5&Q4：How to integrate models with new modalities.
>
> Please refer to the response to Reviewer Z2z7’s Q1.

---

> ### Author Response · Authors · 2024-12-02
> **Looking forward to your feedback!**
>
> Dear Reviewer enUq,
>
> Thank you for your valuable comments and suggestions, which have been very helpful to us. We have carefully addressed the concerns raised in your reviews and included additional experimental results.
>
> As the discussion deadline is approaching, we would greatly appreciate it if you could take some time to provide further feedback on whether our responses adequately address your concerns.
>
> Best regards,
>
> The Authors

---

> ### Author Response · Authors · 2024-12-04
> **Summary of our response**
>
> Dear Reviewer enUq,
>
>
>
> Thank you once again for your valuable suggestions! As the discussion period deadline approaches, we provide a summary of our responses to your concerns.
>
>
>
> **[1. Training and Inference Time]**
>
>
>
> We have detailed the training cost for each stage, showcasing the exceptional efficiency of our approach, which can be completed within a limited time with a single NVIDIA 4090 GPU. For inference, we highlight the significance of increasing the model size to push the boundaries of multimodal representation further.
>
>
>
> **[2.** **Expert Load Balancing]**
>
> We compared the results of applying and not applying the expert load balancing strategy in the MoE structure of LLMs. Based on experimental observations, we discuss why our model does not require additional load balancing mechanisms: different pre-trained expert encoder inherently excels in different input domains. Moreover, we provide detailed router weights for various inputs and analyze the effects of weight routing.
>
>
>
> **[3. Bias between Captions of Different Modalities]**
>
> In the updated manuscript, we include additional feature visualizations to illustrate biases in textual descriptions across different modalities. These distribution gaps of captions for different modalities are evident not only in our model but also in the self-supervised text model, Sentence-T5.
>
>
>
>
>
> **[4. Generalizability to New Modalities]**
>
> We have elaborated extensively on how to bind additional new representation models to enable expansion into new modalities under various scenarios.
>
>
>
>
>
> **[5. Advantage of Item-pair Retrieval]**
>
> We have provided further comparative experiments to demonstrate the effectiveness of item-pair retrieval over the embedding-pair retrieval used in FreeBind. Moreover, we highlight that the greater value of pseudo-item pairs lies in their ability to support the integration of additional data and representation spaces more flexibly—without requiring re-collection of pseudo-embedding pairs during every sapce binding.
>
>
>
>
>
> We sincerely thank you for your thoughtful suggestions. We hope these responses address your concerns. **If you find our replies satisfactory, we kindly ask if you would consider revising your rating based on your updated evaluation of our work.**
>
>
>
> Best regards,
>
>
>
> The Authors

---

### Official Review · Reviewer_qoeh · 2024-11-04

**Soundness:** 4
**Presentation:** 4
**Contribution:** 3
**Rating:** 8
**Confidence:** 3

**Summary:**

This paper presents a novel method for fusing knowledge from mutiple pre-trained spaces across different modalities (including 3D, audio, image, video and language etc.) into a single shared "high-quality omni representation space". The core technique involves a dynamically adjusted weighting schema (achieved via a learnable router) to alleviate interference across knowledge sources/modalities. The lightweight router network that controls binding and routing enables the system to be trained efficiently. The authors performed extensive experiments and demonstrated the superior performance of the approach on multiple relevant benchmarks.

**Strengths:**

* Originality

This paper presents a novel approach for binding cross-modal pre-trained latent space into a different space. Although similar ideas have been published in several prior work, this paper has made some extensions to enable more efficient pseudo item-pair construction and to be able to scale up the target spaces.

* Quality

The effectiveness of the proposed framework is validated by extensive experiments over multiple benchmarks, and the model has achieved even better performance than "individual best" (specialist source model on the target modality only).

* Clarity

The paper is in good shape with key ideas being clearly presented by text and figure illustrations. It is relatively easy to follow.

* Significance

The proposed idea can be easily adapted to new modalities and may lead to more exciting applications.

**Weaknesses:**

* Arguably the idea presented in this paper is a natural extension thus incremental to previous work, e.g. the main change from FreeBind would be the pseudo-item generation (from pseudo-embedding pair aggregation to pseudo-item pair retrieval) and learnable weights routing.

* One missing ablation (correct me if I missed anything) would be a side-by-side comparison of pseudo item pair retrieval vs. pseudo embedding aggregation (e.g. FreeBind), as the authors claim the proposed approach is more efficient and robust. The ablation can be done by applying pseudo item pair retrieval to the same experimental setup as pseudo embedding aggregation if the latter doesn't support more datasets.

**Questions:**

* The foundational step behind the proposed approach is the quality of curated pseudo item-pair across all modalities. Is there any direct or indirect quality measurement of this step? Meanwhile as the results in Table 2 and Table 3 show, after binding the omni representation sometimes beat the original "individual best", suggesting the cross-modality binding can even improve representation capability. It would be interesting to see qualitative examples of such improvement if possible.

* Related to previous question, when a new space needs to be added to the omni representation space, does it require re-computation of all existing spaces? According to equation (4) it seems all embedding spaces need to be adjusted but please provide more clarifications. Also curious if it's possible to iterative do the binding (iteratively do binding -> pseudo item-pair update -> binding etc.) to further boost the performance, if incremental binding (one space at a time) is possible?

* Some minor issues include citations wrong (L53, L60, L124, L155, etc.)

---

> ### Author Response · Authors · 2024-11-25
> **Responses to Reviewer qoeh (1/n)**
>
> ### W1：Core insights of OmniBind.
>
> Our core idea is to explore whether scaling up the binding space can consistently improve multimodal models. These two proposed schemes serve as practical solutions to bring this concept to life. Although we did not incorporate sophisticated designs, we believe **the innovative "scaling up" insight, the practical methods enabling it, and the impressive final performance offer significant value to the community.**
>
>
>
> ### W2： Ablation study of pseudo item pairs and pseudo embedding aggregation.
>
> **We conducted a comparison between pseudo item pairs and pseudo embedding aggregation under the experimental setup of FreeBind.** Specifically, we integrate EVA-CLIP-18B, ImageBind, LAION-CLAP (general), and LAION-CLAP (music). The R@1 accuracy on audio-text, audio-image, and image-text retrieval benchmarks are reported as follows:
>
>
>
> |                     | Audio-Text |           | Audio-Image |           | Image-Text |           |
> | ------------------- | ---------- | --------- | ----------- | --------- | ---------- | --------- |
> |                     | AudioCaps  | Clotho    | VGGSS       | FlickrNet | COCO       | Flickr30k |
> | Pseudo  embedding   | 40.85      | 19.73     | 12.38       | 6.80      | 55.85      | **85.71** |
> | Pseudo  item (ours) | **41.17**  | **20.01** | **12.53**   | **7.20**  | **55.94**  | 85.39     |
>
>
>
> **The results show that pseudo-item pairs generally outperform pseudo-embedding aggregation.** Beyond the empirical performance improvement, as you mentioned, the greater value of pseudo-item pairs lies in their ability to more easily support the integration of additional data and representation spaces.
>
>
>
> ### Q1.1：Quality of curated pseudo item-pair:
>
> **We provide a visualization of pseudo-pair retrieval in the Figure.6,7,8,9 of the updated manuscript.** We can find that the pseudo-pairs obtained from different modalities can effectively capture the unique information of different modalities. In particular, for text descriptions, the text retrieved by different modalities has very obvious different tendencies.
>
>
>
> ### Q1.2: Better performance than "individual best".
>
> Binding the omni representation occasionally outperforms the original "individual best," as observed primarily in 3D classification, 3D-image, and audio-image tasks. Since pseudo items are inherently retrieved using these "individual best" models, we attribute this phenomenon to the enhanced and more discriminative visual and textual representations achieved through the binding process, even when these representations are integrated later into the binding space.
>
>
>
> ### Q2.1：How to integrate new models?
>
> **Details of how to integrate new models are discussed in the response to Reviewer Z2z7’s Q1.**
>
>
>
> Briefly, we bind all representation spaces in parallel to the advanced vision-language model EVA-CLIP-18B, due to its high-quality language and vision representations, and almost all existing multimodal models are either vision- or language-related. **When a new space is bound, we only need to learn a new set of lightweight routers incrementally, and existing learned projectors for binding spaces can be re-used.**
>
>
>
> ### Q2.2 Iterative binding spaces.
>
> Compared with the current parallel binding then weight routing, **iterative binding space will make the pipeline much more complicated** (each time a new space is iteratively bound via a connector, a new router should be trained. In contrast, binding all spaces in parallel only requires training routers one time). **We have been unable to complete the iterative binding experiment these days, while we believe that subsequent research on how to bind spaces more effectively will be an interesting direction.**
>
>
>
> ### Q3: Typos.
>
> Thanks for your reminder, we have rechecked these typos in the updated manuscript.

---

> > ### Comment · Reviewer_qoeh · 2024-11-26
> >
> > Thanks for the additional experiments and explanations! It's a bit unfortunate that the iterative binding experiments are not available yet, as it's a quite important evidence to support the authors claim about "easy-to-expand-to-new-spaces". I've also read the concerns by other reviewers and I agree with some of them. I'll keep my rating for now.

---

> ### Author Response · Authors · 2024-11-27
> **Further Responses to Reviewer qoeh**
>
> Dear Reviewer qoeh:
>
> We are continuing to explore iteratively binding space, as it is more complex than our parallel binding approach. This process will take a few days, and we will update the results as soon as possible.
>
> Thank you for your constructive feedback and further response. We are more than happy to address any further concerns or questions you may have.
>
> Best,
>
> The Authors

---

> > ### Author Response · Authors · 2024-12-02
> > **Further response and thank you for your support!**
> >
> > We conducted experiments on OmniBind-base using two binding strategies: **parallel binding** (as implemented in the paper) and **iterative binding** across each space. Specifically, the iterative binding sequence involves the following steps:
> >
> > 1. Aligning the 3D-Image-Text space with the Image-Text space.
> > 2. Binding the Audio-Image-Text space.
> > 3. Finally, integrate the Audio-Text spaces.
> >
> > The comparative results are summarized below:
> >
> > |                   | Audio-text |           | Audio-Image |          | Image-Text |           | 3D-text   |           |           | 3D-Image  |
> > | ----------------- | ---------- | --------- | ----------- | -------- | ---------- | --------- | --------- | --------- | --------- | --------- |
> > |                   | ACaps      | Clotho    | VGGSS       | FNet     | COCO       | F30K      | Objav.    | Scan.     | Model.    | Objav.    |
> > | Parallel binding  | 43.61      | 20.94     | 14.11       | 7.67     | **56.94**  | **85.99** | **53.30** | **57.79** | **82.82** | 34.34     |
> > | Iterative binding | **44.19**  | **21.73** | **14.33**   | **8.16** | 55.47      | 84.39     | 50.88     | 56.82     | 80.31     | **35.88** |
> >
> > **Overall, both binding strategies achieve comparable performance when using the same knowledge sources.**
> >
> > Specifically, **iterative binding** demonstrates better performance in **audio-text**, **audio-image**, and **3D-image** tasks, while **parallel binding** achieves higher scores in **image-text** and **3D-text** tasks, which is consistent with the strengths and weaknesses of OmniBind compared to individual best. We hypothesize that this discrepancy arises from the **cumulative effect** of iterative binding. Unlike parallel binding, which relies on the individual best to retrieve pseudo data for space binding, iterative binding appears to inherit and amplify both the advantages and limitations of the integrated space relative to the individual best.
> >
> > Finally, we would like to express our sincere gratitude for your thoughtful feedback and support! In our final revision, we will integrate the valuable insights gained through the rebuttal discussions to further enhance the quality of our paper. Thank you once again!
> >
> > Best regards,
> >
> > The Authors

---

### Official Review · Reviewer_ViaF · 2024-11-04

**Soundness:** 3
**Presentation:** 2
**Contribution:** 2
**Rating:** 6
**Confidence:** 4

**Summary:**

The manuscript presents OmniBind, a multimodal joint representation model designed to integrate and optimize the capabilities of 14 pre-trained modalities, including 3D, audio, image, video, and language inputs. By merging these specialized models into a unified representation space, the authors aim to enhance multimodal understanding and generation. The proposed framework employs a dynamic weight assignment mechanism, utilizing lightweight networks to facilitate the binding and routing of knowledge sources. The author claim that they not only improves the overall alignment across different modalities but also allows for effective language representation decoupling.

**Strengths:**

1.The manuscript presents a compelling approach to integrating various multimodal representation models into a unified framework, which is an advancement in the field of human-computer interaction.
2.The introduction of dynamic weight assignment and language representation decoupling may bring benefits.
3.The experimental validation provided in the study partially supports the authors’ claims regarding the versatility and effectiveness of the proposed model.

**Weaknesses:**

1. As for the proposed Language Representation Decoupling, statistics on the number of such cases can be added to support the need for such improvements.
2. Regarding the construction of the pseudo pair data for training, for audio, the authors use audio from AudioSet and then employ a state-of-the-art audio-text model to retrieve the most similar texts from AudioCaps and Clotho (they also retrieve texts from other datasets, but as the authors mention, there is a gap in the texts across different modalities, so the most similar texts are likely to come from the audio-text dataset). First, I think this is somewhat redundant, as they could have directly used the paired audio-text data from AudioCaps, since the audio in AudioCaps comes from AudioSet. Secondly, this means that the training set includes data from Audiocaps, so please confirm whether the methods in Table 2 also have similar situations; otherwise, I believe it would be an unfair comparison. Additionally, the annotations of AudioCaps are highly correlated with the category annotations of AudioSet, which raises concerns about the potential for label leakage. Therefore, I question whether it is appropriate to use AudioSet to validate zero-shot classification in Table 3.
3.The datasets used in the experiments are mostly quite old; it might be worth considering more updated ones.
4.In my opinion the methods used in this paper fall under multi-task learning. Please provide comparisons with multi-task methods to demonstrate the advantages of the approach presented in this paper.

**Questions:**

1. For a certain modality, when different modalities need to be aligned, will it itself require different angles of representation, and is it reasonable to do a universal representation?
2.For language representation decoupling, my understanding is that this involves separately aligning image encoders with text encoders from image-text pretraining, and audio encoders with text encoders from audio-text pretraining, etc. If the authors separate these alignments, does it contradict the goal of achieving a universal representation?

---

> ### Author Response · Authors · 2024-11-25
> **Responses to Reviewer ViaF (1/n)**
>
> ### W1：The necessity of language representation decoupling:
>
> In the newly added Figure 10 of the revised version of our paper, we visualize the representation distribution of text from different domains within different pre-trained models. **Textual descriptions of audio and visual content have inherent domain gaps and should occupy different subspaces in any semantically meaningful space (such as the self-supervised language model, sentence-T5[1]).** Language decoupling reduces the similarity of language representations between audio-related and visual-related modalities from 8.28 to 5.17, reflecting better discernibility and a more dispersed distribution of representations. Additionally, the performance improvements achieved through language decoupling serve as indirect evidence of the enhanced representation quality.
>
> [1] Ni, Jianmo, et al. "Sentence-t5: Scalable sentence encoders from pre-trained text-to-text models." arXiv preprint arXiv:2108.08877 (2021).
>
> ### W2："unfair comparison" and "potential label leakage"
>
> **Our training data strictly adheres to the main baselines, ImageBind++ and InternVL_IB++, and the performance advantage over these methods underscores the effectiveness of our designs.** Additionally, although our data is inherited from previous baselines, **we remove all captions from AudioCaps in the text memory and retrain the OmniBind_Large model to fully eliminate concerns about "potential label leakage."**
>
> |                        | Audio-text |           |           | Audio-Image |          | Image-Text |           | 3D-text   |           |           | 3D-image  |
> | ---------------------- | ---------- | --------- | --------- | ----------- | -------- | ---------- | --------- | --------- | --------- | --------- | --------- |
> |                        | ACaps      | Clotho    | AudioSet  | VGGSS       | FNet     | COCO       | F30K      | Objav.    | Scan.     | Model.    | Objav.    |
> | OmniBind_Large         | **47.89**  | **23.07** | 25.57     | 14.14       | **7.86** | 60.08      | **87.20** | **53.97** | **64.67** | 86.55     | **46.09** |
> | w/o  AudioCaps caption | 46.36      | 22.66     | **27.26** | **14.66**   | 7.68     | **60.08**  | 86.94     | 53.68     | 63.39     | **87.07** | 46.00     |
>
> The results show that there is only a negligible drop in the audio-text retrieval and even stronger zero-shot classification accuracy on AudioSet. **This confirms that AudioCaps captions are not essential for performance and OmniBind's state-of-the-art results are not derived from "potential label leakage."**
>
> ### W3. Experiments on more and latest datasets.
>
> In general, we used the most widely used benchmarks available for each downstream task. Objaverse is the largest and latest test set for 3D-image and 3D classification. AudioCaps and Clotho are the most widely used audio-text benchmarks, and AudioSet is the largest audio classification benchmark, as far as we know. For the audio-image test, we keep the same to the compared baselines. For the image-text task, COCO and Flickr30K are the most commonly used image-text retrieval benchmarks. For video-text retrieval, **we evaluate models on the additional MSVD [2]**. We are in the process of collecting extra latest benchmarks and will share further results shortly. We would be very grateful if you could provide more detailed, latest benchmarks for us to perform comparative experiments.
>
> **Extra results on ImageNet v2 [3] for zero-shot image classification are updated.**
>
> |               | MSVD  | ImageNet v2|
> | ------------- | ----- |  ----- |
> |               | R@1   | Top1  |
> | LanguageBind  | 75.30 | 71.23 |
> | ImageBind     | 69.48 | 69.49 |
> | PointBind     | 69.48 | 69.49 |
> | OmniBind-Full | **77.99** | **73.09** |
> | Individual Best | 79.55 | 77.30 |
>
> [2] Chen, David, and William B. Dolan. "Collecting highly parallel data for paraphrase evaluation." Proceedings of the 49th annual meeting of the association for computational linguistics: human language technologies. 2011.
>
> [3] Recht, Benjamin, et al. "Do imagenet classifiers generalize to imagenet?." International conference on machine learning. PMLR, 2019.

---

> > ### Author Response · Authors · 2024-11-25
> > **Responses to Reviewer ViaF (2/n)**
> >
> > ### W4: Comparisons with multi-task methods.
> >
> > For the research scheme, our method mainly follows the paradigm of FreeBind [3], which aligns different multimodal representation spaces together and is the main baselines we compared in current experiments. The specific research content is multimodal representation, which focuses on mapping different modal inputs into a shared latent space. **We compare our method with another recent multi-task multimodal model, ViT-Len [4].**
> >
> >
> >
> >
> >
> > |                | Audio-text |           | Audio-image |          | Image-text |           | 3D-text   |           |          |
> > | -------------- | ---------- | --------- | ----------- | -------- | ---------- | --------- | --------- | --------- | -------- |
> > |                | ACaps      | Clotho    | VGGSS       | FNet     | COCO       | F30K      | Objav.    | Scan.     | Model.   |
> > | ViT-Lens       | 14.4       | 8.1       | 14.14       | 5.80     | 54.34      | 83.11     | 52.0      | 60.1      | **87.6** |
> > | OmniBind-Base  | 43.61      | 20.94     | 14.11       | 7.67     | 56.94      | 85.99     | 53.30     | 57.79     | 82.82    |
> > | OmniBind-Large | 47.89      | 23.07     | 14.14       | 7.86     | 60.08      | 87.20     | 53.97     | **64.67** | 86.55    |
> > | OmniBind-Full  | **49.28**  | **23.22** | **15.64**   | **8.32** | **63.01**  | **89.12** | **58.13** | 64.60     | 86.83    |
> >
> > The results clearly highlight the strengths of our approach. We hope this can resolve your concerns about "multi-task learning" and we would appreciate it if you could provide more detailed multi-task learning methods for us to perform comparative experiments.
> >
> >
> >
> > [3] Wang, Zehan, et al. "FreeBind: Free Lunch in Unified Multimodal Space via Knowledge Fusion." Forty-first International Conference on Machine Learning.
> >
> > [4] Lei, Weixian, et al. "Vit-lens: Towards omni-modal representations." Proceedings of the IEEE/CVF Conference on Computer Vision and Pattern Recognition. 2024.
> >
> >
> >
> > ### Q1：Significance of universal representations.
> >
> > As far as we know, aligning different modalities in a unified space is a basic issue in multimodal learning and has unique practical significance. Recently, many works focused on this problem, such as Imagebind [5] and CoDi [6], and these prior works extensively discuss the importance of a unified multimodal space. **In simple terms, unified representation for multiple modalities is essential for understanding multiple modalities and generating multimodal content simultaneously (generating arbitrary modalities with arbitrary modalities as conditions), representative works include CoDi [6], Next-GPT[7] and ImageBind-LLM[8].** In addition, we also demonstrated some applications realized by unified representation in the application, such as any-query object localization, any-query audio separation, and complex composable understanding.
> >
> >
> >
> > **Encoding data from the same modality but different domains through multiple perspectives of representation is a key challenge in universal multimodal representation learning.** Our weight routing method focuses on this problem, improving representation discriminability by leveraging dynamic routing. Refer to the response to Reviewer Z2z7’s Q2.1 and Q3 for more discussion about our routing strategy in integrating different semantic biases.
> >
> >
> >
> > [5] Girdhar, Rohit, et al. "Imagebind: One embedding space to bind them all." Proceedings of the IEEE/CVF Conference on Computer Vision and Pattern Recognition. 2023.
> >
> > [6] Tang, Zineng, et al. "Any-to-any generation via composable diffusion." Advances in Neural Information Processing Systems 36 (2024).
> >
> > [7] Wu, Shengqiong, et al. "NExT-GPT: Any-to-Any Multimodal LLM." Forty-first International Conference on Machine Learning.
> >
> > [8] Han, Jiaming, et al. "Imagebind-llm: Multi-modality instruction tuning." arXiv preprint arXiv:2309.03905 (2023).
> >
> >
> >
> > ### Q2: Language decoupling does not contradict universal representations.
> >
> > We want to clarify that **integrating and aligning different pre-trained spaces into a unified space is already achieved in Stage 1 (Binding Spaces)**. After binding space, these **text representations from different pre-trained models have already been mapped to a unified space, which can be directly linearly weighted** (as a counterpoint, it makes no sense to directly add two non-unified text representations). As for Stage 2 (weight routing) where language decoupling is introduced, we are focusing on how to linearly weigh the sub-space within the unified space, and **the result of linearly weighing different representations is still in the same representation space.** Therefore, the language decoupling does not contradict to achieving a universal representation.
> >
> >
> > More analysis of the effectiveness of language decoupling can be found in the response to Reviewer Z2z7’s Q2.1.

---

> > ### Comment · Reviewer_ViaF · 2024-11-30
> >
> > Thanks for the authors' effort in addressing my concerns. I am happy to raise my original ratings to 6.

---

> > > ### Author Response · Authors · 2024-12-02
> > > **Thank you for your support!**
> > >
> > > Dear Reviewer ViaF,
> > >
> > > We appreciate your kind support! In our final revision, we will further improve the paper by incorporating the valuable insights gained from the rebuttal discussions. Thank you again!
> > >
> > > Best regards,
> > >
> > > The Authors

---

### Official Review · Reviewer_Z2z7 · 2024-11-06

**Soundness:** 3
**Presentation:** 2
**Contribution:** 3
**Rating:** 5
**Confidence:** 3

**Summary:**

The paper proposes OmniBind, a framework that aims to merge multiple pre-trained multimodal models into a unified, large-scale representation space that can handle 3D, audio, image, video, and language inputs. This binding strategy employs weight routing to address knowledge interference from multiple sources and optimizes the integration of various models. OmniBind showcases impressive zero-shot generalization and cross-modal alignment across modality pairs.

**Strengths:**

1. This paper effectively addresses the challenge of binding multiple pre-trained multimodal spaces through dynamic weight routing. This approach moves beyond traditional fixed-weight methods by introducing a learnable routing mechanism, enhancing cross-modal alignment and reducing knowledge interference.
2. OmniBind's performance is thoroughly evaluated across diverse benchmarks, covering a wide range of modality combinations. This strengthens the claims of improved generalization and versatility.
3. The paper provides multiple examples of how OmniBind's representation space can be used in various applications, such as any-query object localization and audio separation, which demonstrate the practical benefits of the model.

**Weaknesses:**

1.  While the model uses pseudo-paired data due to a lack of real-world multimodal data pairs, this raises questions about the validity of the results in real-world applications. This reliance on pseudo-pairs could potentially limit the model's robustness in unexpected scenarios.
2. Although the dynamic routing approach is innovative, it may introduce complexity, particularly in practical implementations or deployment, where computational costs and latency might become prohibitive with increasing scale.

**Questions:**

1. If future models introduce new modalities or more granular representations, how easily can OmniBind incorporate them without significant retraining or re-engineering?
2. Given that 3D, audio, image, video, and text often represent different levels of semantic information (e.g., higher ideographic density in text), how does OmniBind address potential biases in multimodal alignment? How did you ensure the reliability and consistency of the pseudo pairs generated across all modalities?
3. Can you provide more details or visualizations on how the routers assign weights in different tasks or datasets? How do these weights correlate with performance on cross-modal benchmarks?

---

> ### Author Response · Authors · 2024-11-25
> **Responses to Reviewer Z2z7 (1/n)**
>
> ### W1： Reliability of pseudo pairs.
>
> First, the pre-trained representation models being integrated together have already been exposed to vast amounts of real-world multimodal data pairs. **Our work mainly tries to connect and combine the different representation spaces, and the knowledge of OmniBind is actually derived from these pre-trained models.** As demonstrated in [1] and [2], **connections between different representation spaces are easy to learn and typically robust, even with limited data.** Lightweight projectors and relatively small datasets can successfully learn the alignment between two representation spaces and exhibit generalization capabilities in downstream tasks.
>
>
> Secondly, our pseudo-paired data distills knowledge from state-of-the-art multimodal models, using their predictions as ground truth. **The correlation knowledge within the pseudo-pairs is also derived from these advanced existing models. In Figure 6,7,8,9 of the updated manuscript, we provide visualization examples of the pseudo pairs**, illustrating the diverse and rich cross-modal alignment information they provide. More importantly, we evaluate our approach on various large-scale benchmark datasets of each modality (Objaverse for 3D, AudioSet for audio, and ImageNet for images). **Across all these benchmarks, our method demonstrates strong generalization performance.**
>
>
>
> Finally, the primary purpose of our pseudo pairs is to construct pairings across all modalities (3D-audio-text-image). **The current implementation, which relies purely on unpaired data, is intended to demonstrate the low data requirements of our approach. In practical usage, we can also incorporate real-world partially paired data.** For instance, retrieve corresponding audio and text for real 3D-image pairs. We hope the potential collaboration with real data can further alleviate the concerns about the robustness of training data.
>
>  [1] Merullo, Jack, et al. "Linearly Mapping from Image to Text Space." The Eleventh International Conference on Learning Representations.
>
> [2] Liu, Haotian, et al. "Visual instruction tuning." Advances in neural information processing systems 36 (2024).
>
> ### W2: Increasing complexity and computational costs.
>
> Integrating multiple models to achieve better performance can be regarded as an effective method to scale up the multimodal representation model, akin to increasing model size for better performance—a common idea in the fields of computer vision (CV) and natural language processing (NLP).
>
> Admittedly, continuously increasing model size raises inference costs. However, inspired by the success of large-scale models in CV and NLP, we believe **scaling up model size and exploring the performance upper bound of multimodal models is a meaningful endeavor, for example, acting as a teacher model for distillation learning (distilling the knowledge of the largest model to small models can foster better models of each size, as DINO v2 [3]).** Proposing larger and more powerful multimodal representation models, therefore, holds substantial practical significance.
>
>  [3] Oquab, Maxime, et al. "Dinov2: Learning robust visual features without supervision." arXiv preprint arXiv:2304.07193 (2023).

---

> ### Author Response · Authors · 2024-11-25
> **Responses to Reviewer Z2z7 (2/n)**
>
> ### Q1: How to integrate future models with new modalities or better representations.
>
> Overall, incorporating new models into OmniBind requires only minimal incremental learning without the need for extensive retraining or re-engineering.
>
> We begin by revisiting our implementation, which consists of two main stages:
>
> Stage 1 (Binding Spaces): we bind all representation spaces to the advanced vision-language model EVA-CLIP-18B in parallel. This model is chosen for its high-quality language and vision representations, and most existing multimodal representation models are correlated with either vision or language.
>
> Stage 2 (Routing Spaces): we select a set of aligned spaces from Stage 1 and learn weight routers to linearly combine these spaces, which were previously bound in parallel.
>
> **Here are detailed ideas to incorporate new models:**
>
> For data, if new models introduce modalities not covered by the current pseudo item pairs, we can utilize overlapping modalities to retrieve additional items in the new modality for the existing pseudo pairs. If no new modalities are introduced, no modifications are required.
>
> For training, in most cases, future models are likely to be correlated with vision or language. In Stage 1, we only need to bind these new models to the basic model, EVA-CLIP-18B. In Stage 2, we train new routers to handle the updated combination of spaces. In rare cases where new models are not directly related to vision or language, we can view the current OmniBind as the basic model. The new models can be integrated by binding them to OmniBind, followed by training routers to weight the new models with the basic model.
>
> In summary, **the existing learned projectors that bind spaces in the current OmniBind can be preserved, and only several extra lightweight MLP projectors are needed, which can be obtained by two training processes.**
>
>
>
> ### Q2.1: Potential biases in multimodal alignment.
>
> **Each pre-trained model inherently reflects specific biases associated with its modality.** For instance, considering audio-text and image-text models, the textual data used during their training often originates from vastly different domains (The newly added Figure 10 in the updated manuscript proves this). This substantial domain gap results in each model exhibiting distinct semantic biases when processing data within the same modality. In OmniBind, the weight router strategy is proposed to integrate the different semantic biases, enabling the inputs from different domains to be processed by appropriate encoders. **More analysis of weight routing in integrating different pre-trained models’ biases is provided in the response to your Q3.**
>
>
>
> ### Q2.2: Reliability and consistency of pseudo pairs.
>
> For the reliability and consistency of the pseudo pairs. First, we use the state-of-the-art retrieval models for each modality pair, which are trained on web-scale data and show impressive robustness. Second, we select each modality as the starting point to retrieve others, respectively, which can effectively expand the diversity of data. The newly added Figure.6,7,8,9 in the updated appendix demonstrates the consistency and diversity of our data. Third, our approach can also accommodate partially-paired real-world data, which may further enhance the reliability in practical usage.

---

> > ### Author Response · Authors · 2024-11-25
> > **Responses to Reviewer Z2z7 (3/n)**
> >
> > ### Q3：Routers assign weights.
> >
> > The router assigns weights of OmniBind-Full in audio-text, image-text, and audio-image tasks and datasets as follows:
> >
> >
> >
> > |                     | Text-audio model |        |          |          |          | Text-image model |          |            |                   |              |           |
> > | ------------------- | ---------------- | ------ | -------- | -------- | -------- | ---------------- | -------- | ---------- | ----------------- | ------------ | --------- |
> > |                     | CLAP1            | CLAP2  | WavCaps1 | WavCaps2 | WavCaps3 | DFN-H14          | EVA-E14p | SigLip-384 | SigLip-so400m-384 | EVA-CLIP-18B | ImageBind |
> > | Text from AudioCaps | 13.67%           | 16.54% | 15.92%   | 14.79%   | 13.55%   | 4.71%            | 3.84%    | 5.63%      | 3.86%             | 3.79%        | 3.72%     |
> > | Text from COCO      | 0.09%            | 0.16%  | 0.12%    | 0.09%    | 0.09%    | 18.56%           | 18.05%   | 19.98%     | 19.32%            | 18.61%       | 4.93%     |
> >
> >
> >
> >
> >
> >
> >
> > |                       | Audio-text  Model |        |          |          |          | Audio-Image  Model |
> > | --------------------- | ----------------- | ------ | -------- | -------- | -------- | ------------------ |
> > |                       | CLAP1             | CLAP2  | WavCaps1 | WavCaps2 | WavCaps3 | ImageBind          |
> > | Audio  from AudioCaps | 13.62%            | 13.60% | 15.05%   | 14.72%   | 14.70%   | 28.32%             |
> > | Audio  from VGGSS     | 13.63%            | 13.47% | 14.66%   | 14.40%   | 14.57%   | 29.27%             |
> >
> >
> >
> > |                   | Image-text  Model |          |            |                   |              | Audio-Image  Model |
> > | ----------------- | ----------------- | -------- | ---------- | ----------------- | ------------ | ------------------ |
> > |                   | DFN-H14           | EVA-E14p | SigLip-384 | SigLip-so400m-384 | EVA-CLIP-18B | ImageBind          |
> > | Image  from COCO  | 13.94%            | 16.56%   | 14.08%     | 14.45%            | 27.01%       | 13.96%             |
> > | Image  from VGGSS | 14.10%            | 19.81%   | 14.31%     | 14.94%            | 22.41%       | 14.43%             |
> >
> >
> >
> > **In summary,** for inputs with significantly different biases (text), our router can dynamically select the appropriate encoders according to the domains of different inputs, thereby effectively integrating the semantic bias of different models and achieving better results. For inputs that do not have specific biases (image and audio), our router maintains overall balance while tending to assign more weight to more powerful representations (such as WavCaps, EVA-CLIP-18B, and ImageBind).

---

> ### Author Response · Authors · 2024-12-02
> **Looking forward to your feedback!**
>
> Dear Reviewer Z2z7,
>
> Thank you for your valuable comments and suggestions, which have been very helpful to us. We have carefully addressed the concerns raised in your reviews and included additional experimental results.
>
> As the discussion deadline is approaching, we would greatly appreciate it if you could take some time to provide further feedback on whether our responses adequately address your concerns.
>
> Best regards,
>
> The Authors

---

> ### Author Response · Authors · 2024-12-04
> **Summary of our response**
>
> Dear Reviewer Z2z7,
>
>
>
> Thank you once again for your valuable suggestions! As the discussion period deadline approaches, we provide a summary of our responses to your main concerns.
>
>
>
> **[1. Reliance on Pseudo-Pairs]**
>
> We first discuss that simple connectors are easy to learn and robust. Next, we provide visualization examples to demonstrate the reliability of pseudo-pairs. Finally, we highlight that our approach does not necessarily require unpaired data; incorporating real datasets can further enhance the reliability of the data in practical applications.
>
>
>
> **[2. Computational Costs and Latency]**
>
> Our core motivation is to explore an efficient and effective way to scale up the multimodal representation model, inspired by the success of scaling approaches in the CV and LLM fields. We highlight the significance of increasing the model size to push the boundaries of multimodal representation further.
>
>
>
> **[3. Integration for Future Models]**
>
> We have elaborated extensively on how to bind future models with better representation or new modalities across various scenarios.
>
>
>
> **[4. Potential Biases in Multimodal Alignment]**
>
> Our weight routing method improves the discrimination of the representation space, enhancing the ability to express different modalities of different information density. Statistical analysis of our router weight assignment also supports this improvement.
>
>
>
> **[5. Statistics of Routers Assign Weights]**
>
>
>
> We provide statistics on the weight assignment by the routers. Two main findings emerge:
>
>
>
> 1) For inputs with significantly different biases (e.g., text), our router can dynamically select the appropriate encoders based on the domains of different inputs, thereby effectively integrating the semantic biases of various models and achieving better results.
> 2) For inputs without specific biases (e.g., image and audio), our router maintains an overall balance while tending to assign more weight to more powerful representations (such as WavCaps, EVA-CLIP-18B, and ImageBind).
>
>
>
> We sincerely thank you for your thoughtful suggestions. We hope these responses address your concerns. **If you find our replies satisfactory, we kindly ask if you would consider revising your rating based on your updated evaluation of our work.**
>
>
>
> Best regards,
>
>
>
> The Authors

---

### Meta-Review · Area_Chair_uA64 · 2024-12-20

**Metareview:**

Given existing multimodal foundation models, this work aims to construct a shared space for those models to support a larger range of modalities, including 3D, audio, image, video and language. The major concerns were insufficient ablation experiments and how to handle new modalities. The rebuttal addressed most of those concerns and Reviewer ViaF and enUq raised the final score to 6. Please incorporate the suggestions and experiments from the discussion in the revised submission. Moreover, please discuss the issue of introduced computational complexity appropriately.

**Additional Comments On Reviewer Discussion:**

After rebuttal, all reviewers found that most major concerns were addressed. Therefore, Reviewer ViaF and enUq raised scores and leaned to accept the work.

---

### Decision · Program_Chairs · 2025-01-22

Accept (Poster)